# Anthropogenic aerosols mask increases in US rainfall by greenhouse gases

Mark D. Risser [1,9] ✉, William D. Collins[2,3,9], Michael F. Wehner [4], Travis A. O'Brien[1,5], Huanping Huang [1,6] & Paul A. Ullrich[7,8]

A comprehensive understanding of human-induced changes to rainfall is essential for water resource management and infrastructure design. However, at regional scales, existing detection and attribution studies are rarely able to conclusively identify human influence on precipitation. Here we show that anthropogenic aerosol and greenhouse gas (GHG) emissions are the primary drivers of precipitation change over the United States. GHG emissions increase mean and extreme precipitation from rain gauge measurements across all seasons, while the decadal-scale effect of global aerosol emissions decreases precipitation. Local aerosol emissions further offset GHG increases in the winter and spring but enhance rainfall during the summer and fall. Our results show that the conflicting literature on historical precipitation trends can be explained by offsetting aerosol and greenhouse gas signals. At the scale of the United States, individual climate models reproduce observed changes but cannot confidently determine whether a given anthropogenic agent has increased or decreased rainfall.

Daily accumulated precipitation, including precipitation associated with extreme events, is an important part of the global water cycle[1]. Precipitation is particularly important considering decreases in natural water storage, including snowpack[2], glaciers[3], and groundwater[4]. As a result, a comprehensive understanding of precipitation change is critical to human systems, including agriculture, water resource management, and infrastructure design. Such knowledge can underpin mitigation policies and adaptation in response to changing risks of natural hazards such as flooding and droughts[5] within a nonstationary global climate[6].

While anthropogenic influence has been identified for many aspects of the Earth system[7–12], robust conclusions regarding the human influence on regional (sub-continental) precipitation remain difficult to obtain. Existing studies primarily address changes at the global scale[13], zonal land-averages[14–16], or continental-scale averages[17–20]. Such large-scale statements about anthropogenic influence on precipitation are highly useful but do not provide the information needed to understand the nature of local climate change, for example, the magnitude and direction (increasing or decreasing) of the change. Attempts to attribute local-scale precipitation trends have proven to be largely inconclusive[21–23] even over the continental United States (CONUS) where there are well-documented century-length trends in seasonal mean and extreme precipitation[24–26]. A recent study[27] identifies a statistically significant human influence on regional precipitation over Europe, but only for mean precipitation and only in winter. One of the primary reasons existing studies struggle to robustly assign human influence is due to their reliance on global climate models, and climate model uncertainty is one of the primary factors that limit confidence in regional attribution for precipitation[28]. For example, anthropogenic aerosols have a significant influence on

[1]Climate and Ecosystem Sciences Division, Lawrence Berkeley National Lab, Berkeley, CA, USA. [2]Earth and Environmental Sciences Area, Lawrence Berkeley National Lab, Berkeley, CA, USA. [3]Department of Earth and Planetary Science, University of California, Berkeley, CA, USA. [4]Applied Mathematics and Computational Research Division, Lawrence Berkeley National Lab, Berkeley, CA, USA. [5]Department of Earth and Atmospheric Sciences, Indiana University, Bloomington, IN, USA. [6]Department of Geography and Anthropology, Louisiana State University, Baton Rouge, LA, USA. [7]Program for Climate Model Diagnosis & Intercomparison, Lawrence Livermore National Laboratory, Livermore, CA, USA. [8]Department of Land, Air, and Water Resources, University of California, Davis, CA, USA. [9]These authors contributed equally: Mark D. Risser and William D. Collins. ✉e-mail: mdrisser@lbl.gov

regional precipitation change over the CONUS[29]; however, the signal-to-noise (SNR) ratio for individual ensemble members of single-forcing anthropogenic aerosol climate model runs ranges from −5 to +3 (indicating that aerosols drive both statistically significant increases and decreases in precipitation; see Supplementary Fig. 1). As such, new methods are needed to attribute human influence on regional precipitation, ideally approaches that reduce direct reliance on global climate models, explicitly model natural drivers of precipitation, disentangle the complex causes of regional precipitation change, and account for anthropogenic aerosols[28,30,31].

The purpose of this study is to implement methods developed for regional detection and attribution (D&A) that provide robust conclusions regarding the human influence on seasonal mean and extreme precipitation over the CONUS. Despite dense measurements of long-term rain gauge records, the Sixth Assessment Report of the Intergovernmental Panel on Climate Change indicates low to medium confidence at best and no agreement at worst in the nature of precipitation change over much of North America [Figure SPM.3;[32]]. Here, we explicitly decompose the uncertain combined anthropogenic signal into separate contributions from two of the most important forcing agents. This decomposition allows us to conclusively attribute changes to these forcing agents. Importantly, our approach utilizes climate model simulations indirectly to identify an appropriate formula for modeling a time series of precipitation [ref. 29, see Eq. (2)] using Pearl-causal inference[33]. Climate model output is then set aside, and we interrogate in situ records from rain gauge measurements which yields Granger-causal[34] attribution statements. Definitive conclusions regarding the spatial patterns and time-to-emergence of human influence on regional precipitation are made possible by simultaneously accounting for both anthropogenic aerosols (globally and locally) and greenhouse gas emissions [much like the two- and three-way analyses in ref. 27]. Complementary analyses of climate simulations assess the degree to which observed relationships can be reproduced by physical models.

Our approach to regional D&A has both similarities and important differences relative to more traditional D&A methods that rely on optimal fingerprinting[35]. A detailed comparison is provided in the Methods (see "Comparison with optimal fingerprinting methods"), but at a high level the important points are as follows. For example, our implementation can be seen as analogous to a two-way regression analysis with single-forcing greenhouse gas-only and anthropogenic aerosol-only experiments. However, unlike optimal fingerprinting, the regressors or so-called "fingerprints" are not model-simulated quantities but instead fixed forcing time series that are reconstructed from observations. The primary motivation of this work is that model uncertainty is a major barrier to regional D&A for precipitation, and hence our methodological choice to not rely on model-simulated responses or inter-model differences is intentional. Also, unlike optimal fingerprinting, we explicitly model part of the internal variability via climate drivers and estimate the magnitude of the internal variability directly from observations, such that uncertainty in this estimate is propagated through to the attribution conclusions. Finally, unlike traditional optimal fingerprinting, we suppose that attribution statements are localized and vary over space (as opposed to a single conclusion for the entire domain), such that D&A statements with uncertainty quantification can be made for either individual grid boxes or spatially-aggregated grid boxes [e.g., all of CONUS or attribution subregions such as in ref. 36] in a single framework. It is also noteworthy that recent research has identified serious problems with the traditional implementation of optimal fingerprinting, namely that it underestimates uncertainty and yields overconfident attribution statements[37,38] and furthermore produces biased estimates of the scaling factors[39].

Over the last century, the principal anthropogenic forcing agents for precipitation over the CONUS are well-mixed greenhouse gases (GHGs) and aerosols[29]. Best estimates of the five GHG concentrations (carbon dioxide, methane, nitrous oxide, and chlorofluorocarbon [CFC] 11 and 12) are available from the boundary condition files used in the sixth phase of the Coupled Model Intercomparison Project (CMIP6)[40] and can be converted to their corresponding radiative forcing on the atmosphere[41,42]; see Supplementary Fig. 2(a). The reconstructed GHG forcing time series used in our analysis involves a time lag to account for the lagged response of sea surface temperatures (SSTs) to GHGs (see the red line in Supplementary Fig. 2d), which dominates the effect of GHGs on precipitation[43,44] (see "Analysis of GHCN in situ records" in Methods). Based on ref. 45, we assume the GHG forcing is spatially uniform across the CONUS.

Anthropogenic aerosols are more difficult to account for since their effects on precipitation are multi-faceted[46] and relevant century-length observed quantities are significantly limited. For example, while GHG forcing imposes primarily a lagged effect (the "slow" precipitation response) on the climate system, the effects of anthropogenic aerosols on precipitation have non-negligible lagged components due to cooling of SSTs as well as fast components due to aerosol-cloud interactions. In order to account for the slow precipitation response to anthropogenic aerosols, we utilize an observationally constrained time series of historical aerosol effective radiative forcing[47,48], denoted "AER-glob"; see Supplementary Fig. 2(c). This forcing time series describes the effects of all non-local anthropogenic aerosols, including remote aerosol emissions from Asia and Europe. As with GHG radiative forcing, we apply a time lag to account for the SST-mediated response on the climate system (see the blue line in Supplementary Fig. 2d). Even though $SO_2$ is the dominant aerosol species for changes in precipitation in the CONUS [Hypothesis 4a of ref. 29, see also Supplementary Table 1], it is nontrivial to explicitly characterize the fast precipitation response to local sulfates in an observational analysis. Without long-term, spatially-resolved observations of, e.g., atmospheric concentrations of $SO_2$, we must rely on climate models. However, the diversity in chemical and physical parameterizations and in atmospheric dynamical formulations across multi-model ensembles yield vastly different concentration and surface deposition rates[49]. Even CONUS-mean $SO_2$-related quantities across simulations from climate models in the Aerosol Chemistry Model Intercomparison Project [AerChemMIP;[50]] differ significantly [see Figure G3 of ref. 29]. Aerosol emissions are a prescribed quantity in historical simulations and are hence consistent across climate models, and ref. 29 show that regionally-averaged time series of $SO_2$ emissions can be used to appropriately quantify the fast precipitation response in each season to anthropogenic aerosols over CONUS. Note that our results are insensitive to the specific method used to derive localized estimates of emissions' influence on precipitation (see Supplementary Fig. 4). Supplementary Fig. 2(b) shows best estimates of CONUS-wide seasonal emissions trajectories from the last century obtained from refs. 51,52, denoted "AER-local". These estimates show that $SO_2$ emissions trend upwards over the first two-thirds of the 20th century (much like GHG forcing) but then, following the introduction of clean air regulations in the mid-1960s, decline sharply to their low present-day levels.

Ultimately, we use the sum-total lagged GHG and AER-glob forcing time series (see the black line in Supplementary Fig. 2d) to quantify the slow precipitation response to anthropogenic influence, and we employ regionally-averaged local $SO_2$ emissions (AER-local) to quantify the fast precipitation response to anthropogenic aerosol forcing. Each of these forcing time series are actually proportional to the fast and slow precipitation response. Statistical attribution coefficients are estimated from rain gauge data to translate the forcing time series to the corresponding rainfall response (see "Analysis of GHCN in situ records" in Methods).

## Results

We analyze in situ measurements of daily precipitation from rain gauges in the Global Historical Climate Network [GHCN;[53,54]], using a set of approximately 2500 high-quality stations with records dating back to 1900. Equipped with the general D&A formula and its simplification for analyzing seasonal mean and extreme daily precipitation over the CONUS presented by [ref. 29, see Eq. (2) in Methods], we generate best estimates of the fast (AER-local) and slow (combined GHG and AER-glob forcing) statistical attribution coefficients, denoted $\beta_{Fast}$ and $\beta_{Slow}$, for $0.25° \times 0.25°$ longitude/latitude grid boxes (see "Analysis of GHCN in situ records" in Methods). Uncertainty quantification allows us to attribute the observed changes to seasonal mean and extreme precipitation for both individual grid boxes and aggregated subregions while accounting for multiplicity in testing (see Methods, ibid.). We further use the attribution coefficients (and their uncertainties) to examine time-to-emergence for each anthropogenic forcing agent (see Methods, "Summarizing the GHCN analysis").

### Spatial scales of attribution: fast versus slow response

The detection and attribution of anthropogenic climate change is inherently a signal-to-noise problem, and a common approach for increasing the signal-to-noise ratio (SNR) is spatial aggregation. Given the general challenges associated with attributing changes to precipitation for individual grid boxes[21] and limited success with attribution for sub-continental scales[22,23,27], our first result examines the spatial scales for which we can confidently attribute changes to regional precipitation using our D&A framework. Starting with the entire CONUS, we subsequently divide the CONUS into two, four, 13, and 75 nested subregions [using the attribution regions defined in ref. 36, see Supplementary Fig. 3 and the right-side panel of Fig. 1] while also assessing individual $0.25° \times 0.25°$ grid boxes. The attribution regions correspond to spatial scales of $\approx 8\,Mm^2$ (all of CONUS), $\approx 4\,Mm^2$ (two subregions), $\approx 2\,Mm^2$ (four subregions), $\approx 0.5\,Mm^2$ (13 subregions), and $\approx 0.1\,Mm^2$ (75 subregions), where $1\,Mm^2 = 1$ million $km^2$; the grid boxes are $\approx 600\,km^2$. For each set of subregions, we area-average the statistical attribution coefficients, test the two subregion-specific null hypotheses $H_{0,f}: \beta_f = 0$ for $f \in \{Slow,Fast\}$ (one for each response), and after applying a multiple testing adjustment, identify the subregions for which we can confidently attribute changes in both the slow and fast precipitation response to external forcing.

Figure 1 tallies the fraction of CONUS for which we can attribute changes with both moderate and strong significance as well as the sign of the attributed change (i.e., if the forcing agent or agents drives increases or decreases in precipitation) across the spatial scales considered. When considering the entire CONUS, we can attribute changes to the fast and slow response with confidence across many seasons, although in some cases attribution can only be made for moderate significance. Note that there is a near monotonic decrease in the fraction of CONUS with significant attribution as we move from large to small scales across forcings, seasons, and precipitation type. For individual grid boxes, the slow response is significant for at least some grid boxes for spring, summer, and autumn for both mean and extreme precipitation. The fast response has a highly significant drying effect for CONUS-average mean precipitation in the winter; in summer and autumn, the fast response instead results in (primarily) enhancements to mean and extreme precipitation, as found in prior multi-model experiments concerning the response of precipitation to sulfates (see below for further detail). In JJA and SON, the enhancements in extreme precipitation are significant for $\approx 0.1\,Mm^2$ (and larger) regions for extreme precipitation and $\approx 0.5\,Mm^2$ (and larger) regions for mean precipitation. These attribution conclusions, which are based on the relative comparison of the signal ($\beta_f$) and the noise (comprised of uncertainty from precipitation vs. $f$ relationships and from internal variability of the climate system), imply that the SNR for the slow precipitation response (AER-glob plus GHG forcing) remains large for

very small spatial scales, while the SNR for the fast response (AER-local) is detectable down to spatial scales of $\approx 0.1$-$0.5\,Mm^2$. Furthermore, since uncertainty due to internal variability of the climate system is the same for all forcing agents *and* the magnitude of the signal is comparable for the fast and slow precipitation response (see Fig. 2), this implies that we have higher certainty for quantifying the slow precipitation response relative to the fast precipitation response at the finest spatial scales considered here.

### Grid-box attribution for precipitation change

Because both GHG and AER forcing have an attributable human influence on mean and extreme precipitation response for individual $0.25° \times 0.25°$ grid boxes in at least one season (Fig. 1), we first explore the statistical attribution coefficients and their significance for GHG, AER-glob, and AER-local forcing at these very small spatial scales. Figure 2 shows the product of the attribution coefficients and the range of each forcing agent such that the plotted units describe the effect of each forcing agent on precipitation (see "Summarizing the GHCN analysis" in Methods); hatching indicates a statistically significant attribution (i.e., that the null hypothesis of no anthropogenic influence is rejected; determined with both moderate and strong significance). The anthropogenic signal is strongest for the GHG influence on 20-year return values of extreme daily precipitation (Fig. 2b), where there is evidence that human-induced GHG forcing causes changes in extremes in all seasons except winter, ranging from 21% of CONUS in summer to 36% of CONUS in spring. GHG forcing primarily causes present-day extreme values to exceed their early-1900 totals by as much as 10 mm day$^{-1}$, with the largest increases in the central US (in winter, spring, and summer), the northern Great Plains (in spring), the southeast (in autumn), and the northeast (in spring and autumn). GHG-driven changes in mean precipitation (Fig. 2a) have similar patterns to those present in extreme precipitation, as evidenced by the high pattern correlation between the effect on mean and extreme precipitation ranging from 0.65 in winter to almost 0.9 in autumn. These relatively strong correlations are noteworthy since changes in means and changes in extremes are driven by different dynamical and thermodynamic mechanisms, such that there is no guarantee of any correspondence in these spatial patterns. Finally, with at least moderate significance we can also conclude that GHGs cause average daily precipitation totals to increase by as much as 1 mm day$^{-1}$ in many places (ranging from 6% of CONUS in spring to 18% of CONUS in autumn). We cannot ascribe statistical significance to grid box mean changes in the winter.

The slow AER response has the opposite sign to that of the slow GHG response by construction (see Methods) since the AER-glob and GHG forcings have opposite signs and the magnitudes of both monotonically increase over most or all of the GHCN record. Therefore significance statements for the slow precipitation response to aerosols are identical to the GHG hatching. As expected from atmospheric theory, the slow precipitation response to aerosols is negative almost everywhere, with only limited areas (and generally non-significant) showing increases to precipitation.

While GHG- and AER-glob-driven increases in precipitation are in line with model-based analyses[55,56], the fast precipitation response to local aerosols (as quantified by $SO_2$ emissions) is much more nuanced. In the winter and spring, increases in $SO_2$ emissions result in drying for both mean and extreme precipitation over 60–75% of the CONUS (see Table 1). This is consistent with the global-mean drying seen in single-aerosol-forcing historical model runs[57]. On the other hand, increased $SO_2$ emissions enhance both mean and extreme precipitation during the summer and autumn. Multi-model mean estimates of the seasonal fast and slow precipitation response to sulfate aerosols (see "Fast versus slow precipitation response to aerosols" in Methods) derived from experiments in the Precipitation Driver and Response Model Intercomparison Project [PDRMIP;[43,57]] reveal that the best estimates of

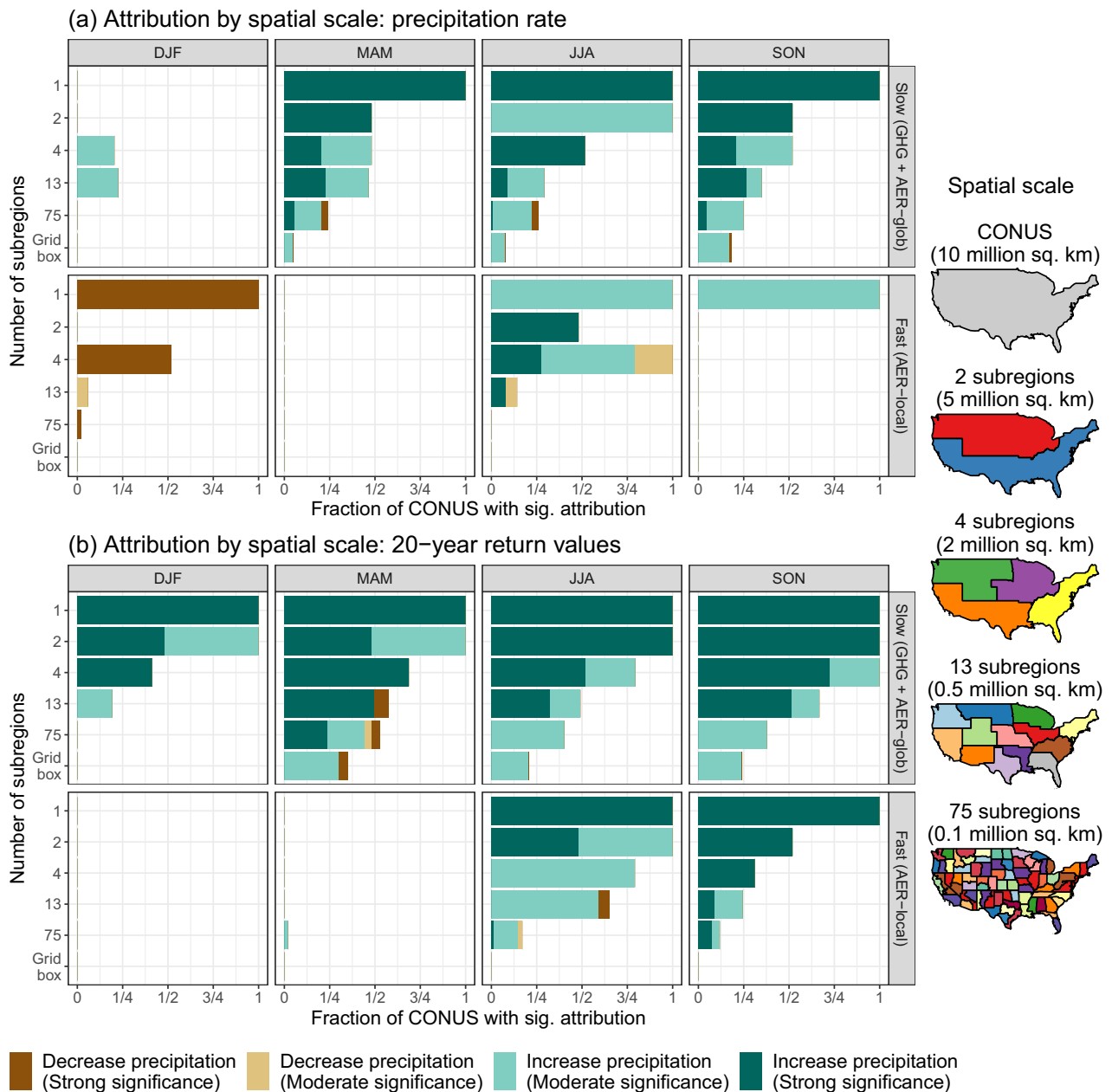

**Fig. 1 | Fraction of the contiguous United States (CONUS) with a significant attribution conclusion for the slow and fast precipitation response in each season across spatial scales. a**, **b** show results for precipitation rate and 20-year return values, respectively. Conclusions are based on null hypothesis tests of no effect for the fast and slow response, and we show results for successively subdividing the CONUS into one, two, four, 13, or 75 regions[36] as well as 0.25° × 0.25° grid boxes. Testing individual subregions or grid boxes accounts for the effect of internal variability, and we include a multiple testing adjustment to yield statistical significance with both moderate and strong significance (see "Methods").

the fast response to aerosols averaged over the CONUS are increases to precipitation, especially in the summer (see Supplementary Fig. 5a). In summer, the spatial patterns of the fast precipitation response to aerosols estimated from PDRMIP (Supplementary Fig. 5b) show strong correspondence with our GHCN-based estimates in Fig. 2: decreases in the eastern U.S. with large and statistically significant increases in the central and northwest U.S. (Supplementary Fig. 6 shows a side-by-side comparison of the spatial patterns).

It is important to note that while our best estimates sometimes indicate $SO_2$ enhancements to both mean and extreme precipitation, in nearly all cases these local changes are often not statistically significant. This lack of statistical significance is because, as mentioned previously, the fast precipitation response to local aerosols is simply much more uncertain than the slow precipitation response.

Lastly, it is clear from Fig. 2 that the the fast and slow precipitation responses are of comparable magnitude, up to ±1 mm day⁻¹ for mean precipitation and ±10 mm day⁻¹ for 20-year return values. Table 1 summarizes the joint distribution of the statistical attribution coefficients for the fast versus slow precipitation response: in winter and spring, it is most common for the slow response (Slow+) to be offset by the fast response (Fast−; for both mean and extreme precipitation), although for roughly 20–30% of the CONUS increases in the slow response are enhanced by additional local AER-driven increases in the fast response (Fast+). For summer and fall, the dominant category is slow response increases further enhanced by fast response increases (again for both mean and extreme precipitation), although the Slow +/Fast − category maintains 20–26% of the domain. The fact that these two anthropogenic agents have an equal effect on precipitation while

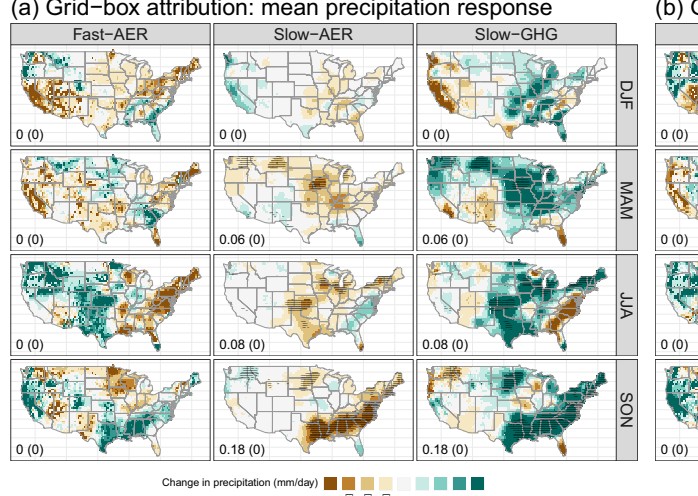

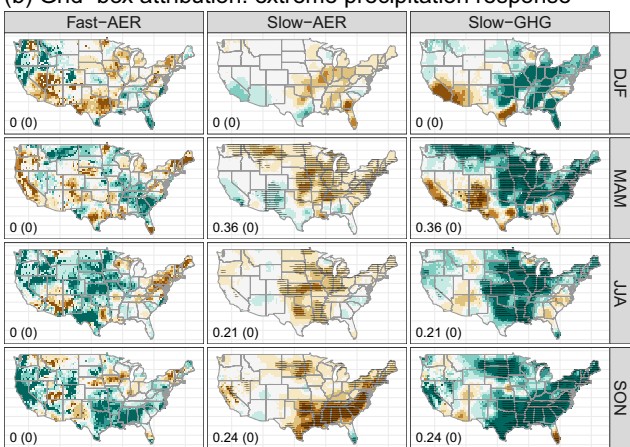

**Fig. 2 | Grid-box attribution for the fast aerosol (Fast-AER), slow aerosol (Slow-AER), and greenhouse gas (Slow-GHG) precipitation responses. a, b** show results for seasonal precipitation rate and 20-year return values, respectively. Hatching indicates where we can attribute a statistically significant human influence, with either moderate (− hatching) or strong (+ hatching) significance. Each subpanel shows the fraction of grid boxes with conclusive attribution at either moderate or strong significance (strong significance only). 20-year return values are calculated as the $1 - \frac{1}{20}$ quantile of the fitted generalized extreme value distribution [71].

both their joint behavior and the fast vs. slow precipitation response to aerosols vary by season reiterates that D&A studies must account for both forcings when attempting to interpret the historical record [as discussed in[28,30]].

### GHG signal emergence masked by aerosols

We next show CONUS-average anomalies for each forcing agent relative to the early 20th century climate in Fig. 3 to assess anthropogenic influence as a function of time. While spatial averaging disguises important heterogeneity in the statistical attribution coefficients (shown in Fig. 2), assessing CONUS-wide averages allows us to evaluate the overall trajectories of precipitation change over the last century. Furthermore, while we are best able to identify anthropogenically-induced changes at the ≈ 8 Mm$^2$ scale (i.e., all of CONUS), we also explore smaller spatial scales later in the section. An important conclusion of ref. 29 was that the total anthropogenic response (denoted ANT) can be represented by summing the individual effects of GHG, AER-glob, and AER-local forcing; hence, we can also compare the CONUS-wide trajectory of the combined anthropogenic influence on precipitation. For both GHG and ANT trajectories we can then identify the year in which each signal emerges (see "Summarizing the GHCN analysis" in Methods; "emergence" is shown by dashed vertical lines in Fig. 3). We do not identify an emergence time for AER-glob or AER-local

because their trajectories are not monotonic over 1900-2020. Note that uncertainties involved in identifying an emergence year in Fig. 3 are different than uncertainties involved in attribution conclusions in Fig. 1: the latter involve a formal hypothesis test for the attribution coefficients (and do not depend on the forcing agents $F_{(\cdot)}(t)$), while the former are a direct function of the forcing agents.

There are three important outcomes regarding the emergence of the isolated GHG signal and combined ANT signal. The first is exemplified by cases where the combined ANT signal emerges very late in the record while the isolated GHG signal emerges much earlier in the record. Thus the emergence of the combined ANT signal is obscured or masked by AER forcing while in fact the expected GHG-induced increases in precipitation have been clear for much of the last half century. This occurs for mean precipitation in autumn and for extreme precipitation in spring and autumn. Second, for mean and extreme precipitation in summer, the GHG-only signal emerges relatively early in the record while the combined ANT signal either never emerges (for mean precipitation) or does not remain above zero by the end of the record (for extreme precipitation). Third, in the two cases where the ANT signal emerges before the GHG signal for mean precipitation in spring and extreme precipitation in winter, it does so only after 2010.

How do issues of aerosol masking the GHG signal play out for spatial scales smaller than ≈8 Mm$^2$ (all of CONUS)? For comparison, we show corresponding time-to-emergence plots for the ≈4Mm$^2$ (dividing CONUS into two subregions) and ≈2Mm$^2$ (dividing CONUS into four subregions) scales in Supplementary Figs. 7 and 8. To summarize these and even smaller spatial scales, where CONUS is divided into 13 subregions, for each season and precipitation type, we identify the emergence time for the isolated GHG signal and sum-total ANT signal as in Fig. 3 for each CONUS subregion and calculate the difference between the two emergence times (ANT minus GHG). These differences are shown in Fig. 4; we also note cases where only one or neither of the GHG/ANT signals emerge. For all spatial scales where either signal emerges, local aerosols mask the GHG signal. This outcome is actually more common for extreme precipitation relative to mean precipitation, indicating that masking from the fast precipitation response to local aerosols is more prominent for extremes. There are

**Table 1 | Area-weighted fraction of CONUS for which the statistical attribution coefficients $\beta_{Slow}$ and $\beta_{Fast}$ are the same sign (positive or negative) or differing sign for each season and precipitation type**

|         |        | Winter | | Spring | | Summer | | Autumn | |
|---------|--------|--------|------|--------|------|--------|------|--------|------|
|         | **Fast** | **−** | **+** | **−** | **+** | **−** | **+** | **−** | **+** |
| **Mean** | Slow − | 0.33 | 0.07 | 0.20 | 0.05 | 0.17 | 0.14 | 0.22 | 0.09 |
|         | Slow + | 0.41 | 0.19 | 0.45 | 0.30 | 0.22 | 0.48 | 0.25 | 0.43 |
| **Extreme** | Slow − | 0.25 | 0.07 | 0.20 | 0.11 | 0.12 | 0.11 | 0.07 | 0.09 |
|         | Slow + | 0.38 | 0.30 | 0.32 | 0.38 | 0.20 | 0.57 | 0.26 | 0.58 |

The "+" symbol indicates that the coefficients are positive, while the "−" symbol indicates that the coefficients are negative.

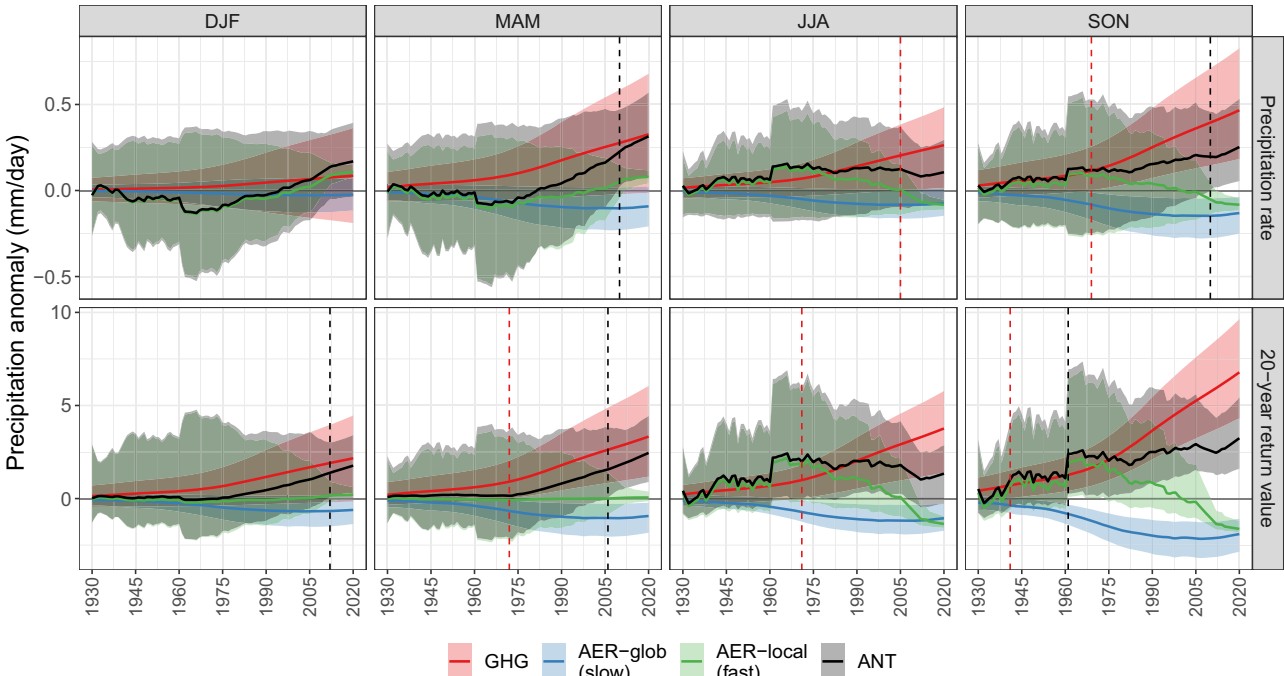

**Fig. 3 | In situ-based, United States-average trajectories of seasonal mean (top) and extreme (bottom) precipitation anomalies from a pre-industrial climate for isolated forcing agents and the combined anthropogenic (ANT) response.** The combined ANT response is the sum of three anthropogenic agents: the slow response from greenhouse gases (GHG; red), the slow response from aerosols (AER-glob; blue), and the fast response from aerosols (AER-local; green). Each trajectory includes a 90% bootstrap confidence band. Dashed vertical lines denote the year of emergence for the isolated GHG signal (red) and combined ANT response (black), defined as the first year in which the 90% confidence band departs from zero and does not return to zero by 2020.

of course some cases where only the ANT signal emerges, but in all cases this occurs at the very end of the record (2010 or later).

In summary, our results show that uncertainties related to the emergence of a detectable and attributable human influence on regional precipitation over the CONUS over the historical record [Figure SPM.3;[32]] can in most cases be explained by decomposing the combined ANT signal into the fast precipitation response to local aerosol forcing and the slow precipitation response to both GHG and AER-glob forcing. Furthermore, we find that the emergence of the isolated GHG signal is masked by local aerosols for spatial scales as small as ≈0.5 Mm².

## Challenges in model-based attribution

Turning to an assessment of global climate models (GCMs), we apply our general D&A formula to individual ensemble members from the CMIP6-historical experiment[58] and apply a weighting scheme that emphasizes internal consistency of historical results relative to pre-industrial and single-forcing runs (see "Analysis of the CMIP6 historical multimodel ensemble" in Methods). Figure 5 summarizes weighted averages of CONUS-wide estimates (including uncertainties) of fast precipitation response to aerosols and the GHG and AER-glob contributions to the slow precipitation response from 316 individual ensemble members representing 25 distinct GCMs. Figure 5 also shows corresponding CONUS-wide estimates from the GHCN rain gauge analysis. As in the previous section, we again focus on CONUS-wide averages for both to simplify presentation and to focus on the outermost scale at which GCMs should have the maximum skill in attributing human-induced changes to precipitation. A large fraction of the multimodel ensemble (MME) captures the expected GHG-driven increases in mean and extreme precipitation, particularly in winter, spring, and autumn. However, it is noteworthy that a non-negligible fraction of the MME cannot rule out GHG-driven drying, particularly for mean and extreme precipitation in the summer. Furthermore, for extremes in the summer, even the best estimate from the average

across the MME suggests that GHG forcing results in drying. The slow precipitation response to aerosols (summarized by AER-glob forcing) is generally drying, although the effect on precipitation is generally smaller for Slow-AER versus Slow-GHG since the maximum forcing range is reduced (−0.89 W m⁻² for AER-glob versus +2.61 W m⁻² for GHG, see Supplementary Fig. 2).

A similar result holds for the MME estimates of the fast precipitation response to aerosols, where large numbers of ensemble members suggest that increased $SO_2$ emissions lead to either increases or decreases in mean and extreme precipitation. The station-data-based 90% confidence intervals shown in Fig. 5 have at least some overlap with the central 90% of the weighted CMIP6 MME across all seasons, precipitation type (mean and extreme), and anthropogenic forcing agent. This implies that climate models are consistent with GHCN-based estimates, although the model uncertainty is so high that the sign of the trend cannot be discerned (whereas it can for GHCN due to lower uncertainty). The degree of consistency between climate models and observations is of course differentiated, with higher consistency for GHG-driven changes in extreme precipitation in winter and spring than local AER-driven changes in extreme summer precipitation.

Our analysis of the large CMIP6 MME reiterates that if one is to pursue D&A of regional precipitation with climate models (a pursuit we advise against with current models), it is important to use multiple climate models. The weighted summaries of individual ensemble members in Fig. 5 illustrate the broad range in GCM-simulated responses to anthropogenic forcing, meaning that individual climate models cannot confidently attribute even the sign of the effect of AER and GHG forcing on CONUS precipitation up through present day. This is particularly true for aerosols, although (as previously mentioned) in some cases individual ensemble members indicate that increases to GHG forcing result in drying. The implication is that D&A for regional precipitation based on a single-model can yield opposite results relative to identical assessments using other climate models in the MME,

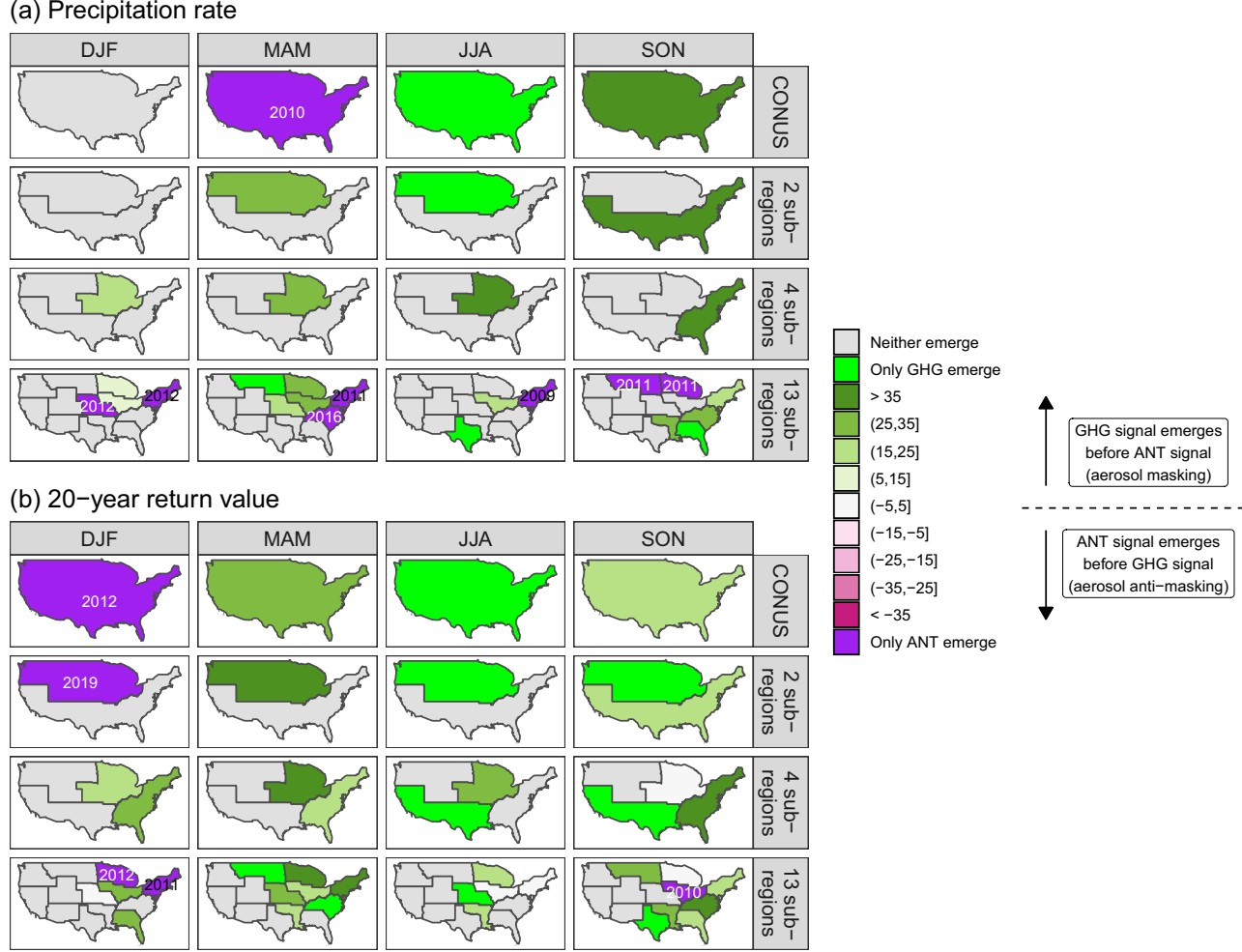

**Fig. 4 | Differences in emergence times for the combined anthropogenic response (ANT) versus the isolated greenhouse gas (GHG) signal across spatial scales. a, b** show results for seasonal precipitation rate and 20-year return values, respectively. The plotted color represents the difference in the year of ANT emergence minus the year of GHG emergence; green colors indicate masking from local aerosols while pink and purple colors indicate anti-masking from local aerosols. For cases where only the ANT signal emerges, we show the year in which the ANT signal emerges in black and white text.

including GHG-driven drying and AER-driven moistening (sometimes "conclusively"; see Supplementary Fig. 9). Reassuringly, the weighted CMIP6 multimodel ensemble means are much more consistent with the observations (also shown in Fig. 5), although the ensemble means appear to be overconfident in their assessment of uncertainty [as has been observed elsewhere, e.g.,[59]]. A notable exception is in forced changes to summer precipitation, particularly for GHG-driven changes to extremes, where the multimodel mean indicates, seemingly erroneously, high confidence in GHG-induced CONUS-wide drying rather than moistening.

## Discussion

The combination of large observational uncertainty, model uncertainty, and internal variability have made it difficult for traditional D&A methods to obtain conclusive statements regarding the human influence on regional precipitation change. Here, we have explicitly quantified the composition of forced precipitation changes over the United States by developing and implementing methods that use model simulations offline from observational analysis and simultaneously account for multiple anthropogenic agents. We anticipate that the results in this paper, which provide Granger-causal statements[34], will provide a foundation for more traditional Pearl-causal D&A studies[33], much in the same way that Risser and Wehner (2017)[60] provided motivation for Patricola and Wehner (2018)[61].

The assumptions underlying Eq. (2) are strictly limited to the historical period, and therefore we cannot extrapolate our results to compare with, e.g., CMIP-based projections. However, the CMIP6 analysis summarized in Fig. 5 can be used to compare GCM-based spatial patterns of the sum-total forced change over 1900-2014 (the period for which our assumptions are justified) with corresponding observational quantities. Supplementary Fig. 12 shows the spatial patterns of change for 2014 forcing conditions (GHG and $SO_2$ emissions) versus 1900 levels, both for the GHCN data and also the weighted multimodel mean (stippling for these figures now indicates where the changes are indistinguishable from zero). As with the multimodel mean results shown in Fig. 5, the GCM-based spatial patterns appear to be overconfident, but otherwise show relatively good agreement with the observations (again except for summer changes). This result is both reassuring and in line with previous examples of the multimodel mean yielding much better agreement with observations than any individual ensemble member or model[62].

Our conclusions underscore the importance of considering both century-length records and multiple anthropogenic forcing agents when calculating observed trends and conducting regional D&A analyses. For example, Fig. 3 shows that at the scale of CONUS, the overall trajectories of mean and extreme precipitation in the summer and autumn are essentially flat from 1960-2020 (despite ANT emergence early in the record for extremes) while winter means and extremes

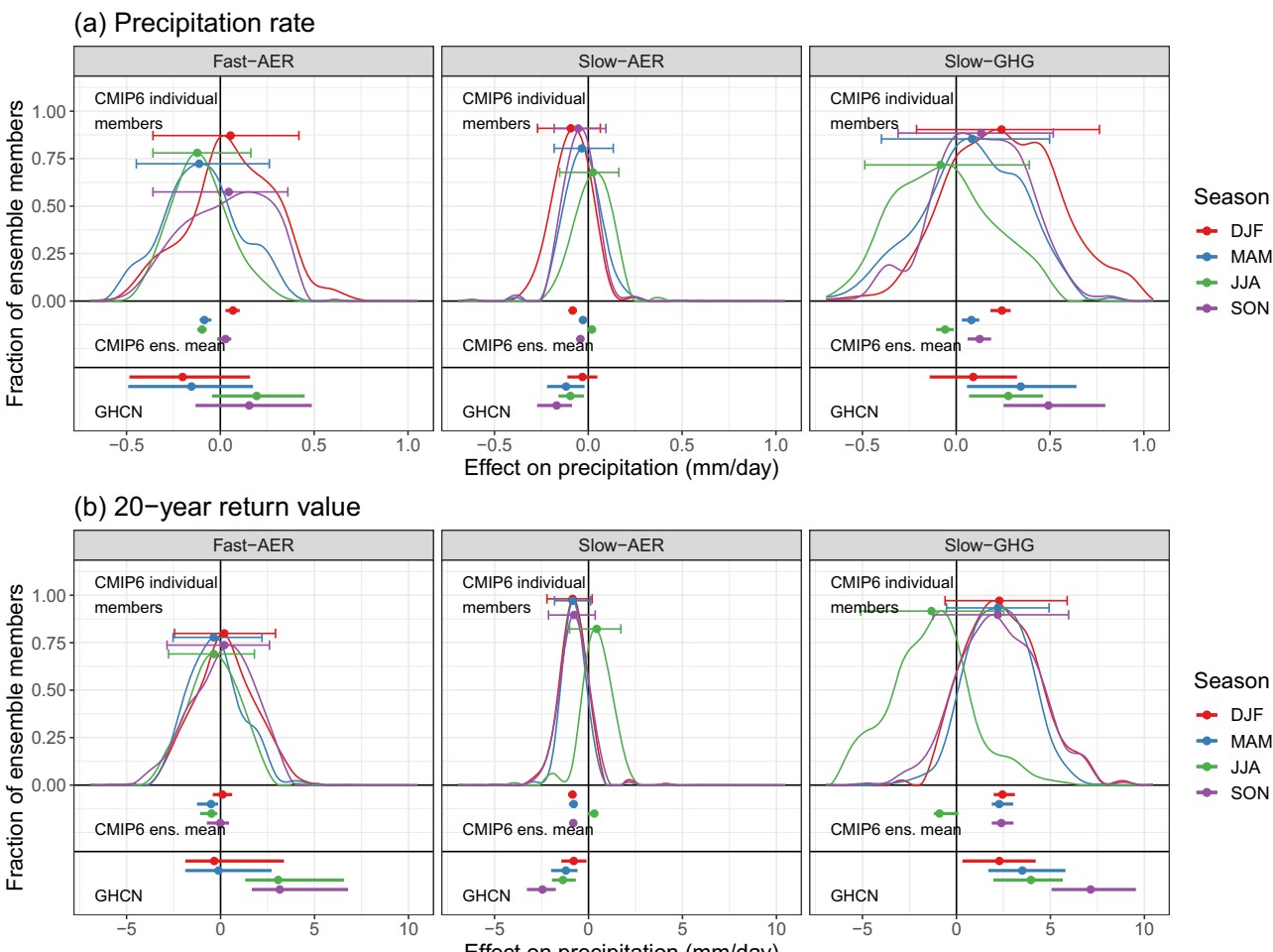

**Fig. 5 | Comparison of the United States-average effect of each forcing agent on precipitation for historical Coupled Model Intercomparison Project Phase 6 (CMIP6) simulations and rain gauge measurements from the Global Historical Climate Network. a**, **b** show results for seasonal precipitation rate and 20-year return values, respectively, and we show both individual CMIP6-historical ensemble members and the ensemble mean. Estimates compare maximum versus minimum levels of each forcing agent (see "Methods"), and the *y*-axis shows the fraction of the multi-model ensemble that is consistent with each value on the *x*-axis. CMIP6 estimates involve weights based on the internal consistency of each global climate model (see "Methods"), and error bars overlaid on each density summarize the central 90% and median area under each curve.

show strong increases over 1960-2020 (despite there being no overall ANT signal for winter means). This shows that our results are actually quite consistent with, e.g., [ref. 63, see their Figure 10], who found that over 1980–2020 the risk of rainfall extremes is unchanging in the summer and fall but increasing in the winter. Considering regional precipitation trends, it is notable that the peak effect of $SO_2$ emissions on mean summer precipitation (Fig. 2a) is approximately 0.7 mm day$^{-1}$ in the Pacific Northwest while the effect of GHG forcing is negligible in the same area. Since forcing due to $SO_2$ emissions peaked in 1966 and has been declining since, this implies that a simple time series analysis of mean precipitation in this region for the last half of the 20th century (e.g., from 1950 onward) might show a *negative* trend. This example demonstrates that our results provide an important foundation for understanding regional trends in precipitation (see also Supplementary Figs. 7 and 8).

Throughout the scientific literature on D&A, "human-induced" or "anthropogenic" is used to refer to a combined anthropogenic effect, including GHG emissions, aerosols, and other human influences taken together. An important contribution of this paper is our ability to use in situ observations to explicitly identify and isolate how the anthropogenic signal is composed of both GHG and AER forcing. Recall that we previously showed that other sources of external forcing do not have a meaningful effect on seasonal precipitation over the CONUS,

including land use/land cover change, natural forcings (including solar and volcanoes), and stratospheric ozone[29]. While uncertainties regarding the combined anthropogenic influence on regional precipitation summarized in, e.g., Figure SPM.3 of ref. 32 remain, our ability to isolate the GHG-only trajectory is nonetheless useful from a communications standpoint. For example, we posit that the public interprets the phrase "human-induced climate change" as primarily involving GHG emissions; hence, it is useful to highlight that the combined anthropogenic signal would have emerged (in some cases) much earlier if not for the counteracting (in some cases) effects of AER forcing. Understanding the relative contributions of individual forcing agents, particularly GHG forcing, is also highly useful for characterizing different scenarios of future projections.

While the analyses in this paper focus on precipitation, our results contribute to mounting evidence of GHG-driven increases in flood risk, particularly in the western United States. A recent study by ref. 64 shows that increases in flood hazard have been "masked" over the past fifty years by internal variability, which agrees with the more general conclusion in ref. 65 regarding the amplification of internal variability under climate change. Ultimately, a combination of the increased moisture-holding capacity of the atmosphere forced by GHGs, decreased masking by anthropogenic aerosols, and an amplification of internal variability from large-scale warming points toward dramatic

increases in flood risk in the near future. As described in ref. 66, these increases will negatively affect many aspects of human society.

## Methods

### Analysis of GHCN in situ records

The central results of this paper are based on an analysis of in situ measurements of daily precipitation from the GHCN-daily database [GHCN;[53,54]]. Specifically, we use measurements from a set of $n = 2480$ high-quality weather station records in the contiguous United States (CONUS) that have at least 66.7% non-missing daily values over December 1, 1899 to November 30, 2020. This set of stations is a subset of the more than 21,000 total GHCN stations in the CONUS; the geographic distribution of the high-quality stations is shown in Supplementary Fig. 11. For each station, we calculate seasonal mean precipitation rates (mm day$^{-1}$) and seasonal maximum daily precipitation rates (mm day$^{-1}$, often referred to as Rx1Day) in each year so long as there are no more than 33.3% missing measurements in the season-year[29]. Developed a general framework for modeling a time series of seasonal mean or maximum daily precipitation $\{P(t)\}$, where the temporal units $t$ refer to a year. For simplicity, we suppress notation for geographic location; we later apply this formula to seasonal precipitation from each of the GHCN gauged locations. Our approach statistically models precipitation as

$$P(t) = \underbrace{P_0}_{\text{Pre-ind.}} + \underbrace{P_F(t)}_{\text{Forced}} + \underbrace{\underbrace{P_D(t)}_{\text{Low-freq. Drivers}} + \underbrace{P_W(t)}_{\text{Weather}}}_{\text{Internal variability}} \tag{1}$$

where the time series is decomposed into a pre-industrial component $P_0$ that represents an overall average uninfluenced by external forcing, an externally forced component $P_F(\cdot)$, and an internal variability component comprised of a "driven" term $P_D(\cdot)$ that describes year-to-year variability due to known modes of large-scale oceanic and atmospheric drivers together with a term $P_W(\cdot)$ associated with short-term weather variability. One strength of this framework is that it explicitly characterizes part of the internal variability via $P_D(\cdot)$ [as advocated by ref. 28]. The various analyses in ref. 29 (summarized in Supplementary Table 1) verified that the additive framework of Eq. (1) is appropriate and can safely be simplified when considering seasonal precipitation over the CONUS in the historical record (1900-present) as follows:

$$P_F(t) \approx \beta_{\text{Slow}} F_{\text{Slow}}(t, \tau_{\text{Slow}}) + \beta_{\text{Fast}} F_{\text{Fast}}(t, \tau_{\text{Fast}}),$$
$$P_D(t) \approx \sum_{d = \mathcal{D}} \beta_d \, d(t),$$
$$\text{Var} \, P_W(t) = \sigma^2(t) \approx V_0 \times \exp\{V_1 t\}, \tag{2}$$
$$1 - \frac{\text{Var} \, P_W(t)}{\text{Var} \, P(t)} \approx Constant.$$

Historically, the principal anthropogenic forcing agents for CONUS precipitation are well-mixed greenhouse gases (GHGs) and aerosols (AER), which collectively define a fast and slow precipitation response:

$$F_{\text{Slow}}(t, \tau_{\text{Slow}}) = F_{\text{GHG}}(t, \tau_{\text{Slow}}) + F_{\text{AER-glob}}(t, \tau_{\text{Slow}}),$$
$$F_{\text{Fast}}(t, \tau_{\text{Fast}}) = F_{\text{AER-local}}(t, \tau_{\text{Fast}}), \tag{3}$$
$$\tau_{\text{Slow}} = 14, \tau_{\text{Fast}} = 0.$$

Here, $F_{\text{GHG}}(t, \tau_{\text{Slow}})$ is the lagged GHG forcing time series shown in Supplementary Fig. 2(a), $F_{\text{AER-glob}}(t, \tau_{\text{Slow}})$ is the lagged aerosol effective radiative forcing [which applies globally, hence "AER-glob";[47]] shown in Supplementary Fig. 2(c), and $F_{\text{AER-local}}(t, \tau_{\text{Fast}})$ is the regionalized (local, hence "AER-local") $SO_2$ emissions. All forcing time series are reconstructed from observations and considered fixed. Equation (2) specifies that the forced component can then be described by a linear sum of a coefficient $\beta_{(\cdot)}$ multiplied by the forcing

time series $F_{(\cdot)}$. We henceforth denote $\beta_{\text{Slow}}$ and $\beta_{\text{Fast}}$ as the "statistical attribution coefficients," since in our approach these are the analog of scaling factors in an optimal fingerprinting analysis, e.g., ref. 35 (see below for additional details). Furthermore, note that the attribution coefficients (which are estimated from data) translate the fixed forcing time series into the corresponding fast and slow precipitation response. Following ref. 29, we set $\tau_{\text{Fast}} = 0$ years, and for the GHCN analysis we use $\tau_{\text{Slow}} = 14$ years [the CMIP6 multimodel ensemble average of the lagged response to GHG forcing; see Appendix A of[29]]. Note that we use stochastically regionalized (i.e., spatially-varying) $SO_2$ emissions data to characterize $F_{\text{AER-local}}(t, \tau_{\text{Fast}})$; again see ref. 29. Following ref. 67, the driven component $P_D(t)$ can be well-approximated by a linear function of climate drivers $\mathcal{D}$, comprised of the El Niño Southern Oscillation (ENSO) Longitude Index (ELI), the Arctic Oscillation (AO), the North Atlantic Oscillation (NAO), the Pacific-North American pattern (PNA), and the Atlantic Multidecadal Oscillation (AMO). Note that, as described in ref. 67, the AO is excluded from the December/January/February (DJF) analysis due to its strong coupling (and high correlation) with the NAO in this season. The residual weather variability term $P_W(t)$ is modeled statistically as following either a Gaussian distribution for the seasonal mean (with mean zero and variance $\sigma^2(t)$) or a Generalized Extreme Value distribution for the seasonal maxima (centered on zero with variability described by $\sigma^2(t)$ and time-invariant shape parameter). The seasonal time series of the drivers for the GHCN analysis are calculated from various observational products; see Section 2 of ref. 67. The variability of $P_W(t)$ is modeled as the product of a "baseline" or pre-industrial variance $V_0$ and a time-varying quantity $\exp\{V_1 t\}$, but the $P_W(\cdot)$ are otherwise statistically independent (i.e., we assume that all autocorrelation in the time series is fully captured by $P_F(\cdot)$ and $P_D(\cdot)$). Finally ref. 29, shows that the signal-to-noise ratio of the time series is approximately constant.

Our analysis of the GHCN records proceeds in three steps, following the methodology developed in refs. 68 and 26. First, independently at each station, we obtain maximum likelihood estimates of all unknown statistical parameters in Eq. (2), including the statistical attribution coefficients $\beta_{\text{Slow}}$ and $\beta_{\text{Fast}}$. Second, utilizing all stations, we apply a spatial statistical approach using second-order nonstationary Gaussian processes to interpolate each statistical parameter to obtain best estimates of each field for a high-resolution $0.25° \times 0.25°$ longitude-latitude grid over CONUS. Third, we quantify uncertainty via resampling methods, specifically a block bootstrap for estimating standard errors and confidence intervals [as in ref. 68] and a permutation/reshuffling approach for ascribing statistical significance to spatial patterns [as in ref. 26]. Once we have the best estimates and uncertainty quantification for the statistical attribution coefficients, the attribution exercise proceeds by testing the null hypothesis $H_{0,f} : \beta_f = 0, f \in \{\text{Slow, Fast}\}$ for either individual grid boxes or spatially-aggregated grid boxes (e.g., the nested attribution regions defined in[36]). Rejecting $H_{0,f}$ implies that there is a significant relationship between forcing agent(s) $f$ and seasonal precipitation, i.e., we can conclusively attribute changes in precipitation to the human activity described by agent $f$. Note that for a given spatial location we can only attribute the sum-total GHG and AER-glob slow precipitation response (via statistical inference on $\beta_{\text{Slow}}$); however, we can separate the individual effect of GHG and AER-glob forcing on precipitation over time when assessing emergence times (see "Summarizing the GHCN analysis" below). As in refs. 26,67, the statistical significance of each $H_{0,f} : \beta_f = 0$ is determined with both "moderate" and "strong" significance; furthermore, when a set of simultaneous tests is conducted (e.g., for a set of CONUS subregions) we apply a multiple testing adjustment that bounds the proportion of type I errors at 0.33 (for moderate significance) and 0.1 (for strong significance).

## Comparison with optimal fingerprinting methods

In contrast with our approach, traditional D&A methods rely on optimal fingerprinting [OF;[35]], which regresses an observed quantity onto a linear combination of model-simulated responses to a set of external forcings. OF-based analyses comprise the large majority of attribution statements cited in the IPCC but, as mentioned in the introduction, have yielded largely inconclusive results for attributing changes to regional precipitation. The OF approach proposes the following statistical model:

$$\mathbf{Y} = \sum_{f \in \mathcal{F}} \beta_f (\mathbf{F}_f - \mathbf{e}_f) + \mathbf{e}_0 \qquad (4)$$

where $\mathbf{Y}$ represents an observed quantity, $\mathcal{F}$ is a set of relevant experiments involving one or more external forcings, the $\mathbf{F}_f$ are corresponding model-simulated quantity from experiment $f$ (often called the "fingerprints" and typically representing some sort of model ensemble average), $\mathbf{e}_f$ represents sampling uncertainty in the model-simulated $\mathbf{F}_f$, $\mathbf{e}_0$ quantifies the effect of internal variability on the observations, and the $\beta_f$ are scaling factors used to make causal statements. Specifically, a fingerprint $\mathbf{F}_f$ is "detected" in the observations if the uncertainty limits on $\beta_f$ do not include zero, and $\mathbf{F}_f$ is furthermore attributable if the uncertainty limits on $\beta_f$ are consistent with one. In Eq. (4), $\mathbf{Y}$ and the $\mathbf{F}_f$ are typically vectors representing values indexed over space and/or time as anomalies from some baseline period. Typical applications explore either so-called "one-way" regression analysis wherein $\| \mathcal{F} \| = 1$ (considering only an all-forcings experiment or only a single-forcing experiment) or "two-way" regression analysis with $\| \mathcal{F} \| = 2$ where one attempts to separate the total anthropogenic and natural responses [using two single- or multiple-forcing experiments; see, e.g., ref. 27].

  Our approach (Eq. (2)) can be seen as a special case of Eq. (4). First, we implement a two-way regression where we consider two individual forcing agents, namely GHGs and AER, separated into fast and slow response components; i.e., $\mathcal{F} = \{ \text{Slow}, \text{Fast} \}$, where Slow = GHG + AER-glob and Fast = AER-local. Recall that[29] explicitly showed that a two-way regression is appropriate by verifying that (1) including only GHG and AER is sufficient for D&A over CONUS and all other external forcings are negligible; and (2) the total anthropogenic response can be represented by the sum of GHG and AER forcing (global and local). The observed vector in Eq. (4) represents yearly measurements of seasonal mean and maximum precipitation over time from a single geospatial location, i.e., $\mathbf{Y} = \{ P(t) : t = 1, \ldots, T \}$. Unlike Eq. (4), in our implementation the $\mathbf{F}_i$ are not model-simulated precipitation but instead fixed forcing time series that are reconstructed from observations. The mechanisms for how GHG forcing impacts precipitation are relatively well understood and the forcing time series $F_{\text{GHG}}(\cdot)$ has relatively low uncertainty; however, unlike GHG forcing, the AER-glob forcing time series has non-negligible uncertainty, such that we must account for non-zero $\mathbf{e}_{\text{Slow}}$. Similarly, while the $SO_2$ emissions are well-observed, their influence on precipitation is less certain, making it important to also account for non-zero $\mathbf{e}_{\text{Fast}}$. Both sources of uncertainty are handled in a Monte Carlo sense via (1) probability-weighted trajectories of AER-glob forcing for $\mathbf{e}_{\text{Slow}}$ [the 5th, 16th, 50th, 84th, and 95th percentile trajectories; see Supplementary Fig. 2 and[47,48]] and (2) an ensemble of regionalized emissions for $\mathbf{e}_{\text{Fast}}$[29]. Since the units of the $\mathbf{F}_f$ are no longer mm day$^{-1}$, the $\beta_f$ are no longer unitless (as they are in Eq. (4)) and instead have units mm day$^{-1}$ per unit increase in the forcing time series. Therefore, in our case the D&A exercise is reduced to assessing a single null hypothesis test (i.e., rejecting $H_{0,f} : \beta_f = 0$ both detects and attributes forcing(s) $f$), such that larger $\beta_f$ (in absolute value) implies a stronger influence of forcing $f$. An important feature of our implementation is that we partially model the internal variability via the driven component, as advocated in ref. 28; while the elements of $\mathbf{e}_0 = [e_0(1), \ldots, e_0(T)]$ describe all non-externally-

forced variability, in our approach the non-forced variability is explained by both the driven term $P_D(t)$ and the weather variability term $P_W(t)$. In other words, one can relate $e_0(t) = P_D(t) + P_W(t)$, such that by definition we have $\text{Var } P_W(t) \le \text{Var } e_0(t)$. Since uncertainty in $\beta_f$ is largely a function of the magnitude of internal variability (here $\text{Var } P_W(t)$ vs. $\text{Var } e_0(t)$), our approach will increase the signal-to-noise ratio relative to a more traditional approach. Unlike Eq. (4), where the variance-covariance matrix of $\mathbf{e}_0$ is estimated offline from a set of pre-industrial control runs, the variance-covariance matrix of the weather variability $P_W(t)$ is estimated using only the observations and simultaneously with the $\beta_f$. Finally, our methodology proposes that the $\beta_f$ is not a scalar quantity but instead a spatial process, i.e., varying across the spatial domain of interest. Critically, this allows us to estimate different $\beta_f$ values for a high-resolution $0.25° \times 0.25°$ grid, such that D&A statements can be made for either individual grid boxes or aggregated grid boxes in a single framework. In other words, it is trivial to generate spatially aggregated D&A statements across various spatial scales.

## Summarizing the GHCN analysis

While the results presented in this paper focus on the statistical attribution coefficients from the forced component, i.e., $\beta_{\text{Slow}}$ and $\beta_{\text{Fast}}$, for the sake of interpretability and cross-comparison we convert each of these values to an effect on precipitation. These summaries directly account for uncertainty in the aerosol forcing time series $F_{\text{AER-glob}}$ and $F_{\text{AER-local}}$. For each season and precipitation type, the relevant output of our statistical analysis are estimates of the attribution coefficients

$$\left\{ \widehat{\beta}^{a,r}_{\text{Slow}}(g), \widehat{\beta}^{a,r}_{\text{Fast}}(g) : a = 1, \ldots, 5; r = 1, \ldots, 100 \right\} \qquad (5)$$

at each grid cell $g$, for AER-glob trajectory $a = 1, \ldots, 5$ and regionalized $SO_2$ emissions trajectory $r = 1, \ldots, 100$. (Note that there are corresponding estimates for bootstrap and permutation resampling; these are used to quantify uncertainty.) For the "Slow-AER" (AER-glob) maps shown in Fig. 2, each grid cell shows the best estimates

$$\Delta \widehat{P}^{\text{GHCN}}_{\text{AER-glob}}(g) = \frac{1}{100} \frac{1}{\sum_a w_a} \sum_{a=1}^{5} w_a \Delta^a_{\text{AER-glob}} \sum_{r=1}^{100} \widehat{\beta}^{a,r}_{\text{Slow}}(g), \qquad (6)$$

where $\Delta^a_{\text{AER-glob}}$ is the change in lagged AER-glob forcing for 1900 vs. 2010 with $\tau_{\text{Slow}} = 14$ years for each of the five trajectories shown in Supplementary Fig. 2, the $w_a = \phi(p_a)$ are weights derived from the standard Normal probability density function (where $\{p_a\} = (0.05, 0.16, 0.5, 0.84, 0.95)$), and $\widehat{\beta}_{\text{Slow}}(g)$ are from Eq. (5). The reference years 1900 and 2010 are chosen since these are when AER-glob forcing is at its minimum and maximum, respectively, in the time period we are analyzing. For the "Slow-GHG" maps shown in Fig. 2, each grid cell shows the best estimates

$$\Delta \widehat{P}^{\text{GHCN}}_{\text{GHG}}(g) = \Delta_{\text{GHG}} \frac{1}{100} \frac{1}{\sum_a w_a} \sum_{a=1}^{5} w_a \sum_{r=1}^{100} \widehat{\beta}^{a,r}_{\text{Slow}}(g), \qquad (7)$$

where $\Delta_{\text{GHG}} = 2.61$ W m$^{-2}$ is the change in lagged GHG forcing for 1900 vs. 2020 with $\tau_{\text{Slow}} = 14$ years and $\widehat{\beta}_{\text{Slow}}(g)$ are from Eq. (5). The reference years 1900 and 2020 are chosen since these are when GHG forcing is at its minimum and maximum, respectively, in the time period we are analyzing. For the "Fast-AER" (AER-local) maps, each grid cell shows

$$\Delta \widehat{P}^{\text{GHCN}}_{\text{AER-local}}(g) = \frac{1}{100} \frac{1}{\sum_a w_a} \sum_{r=1}^{100} \Delta^r_{\text{AER-local}} \sum_{a=1}^{5} w_a \widehat{\beta}^{a,r}_{\text{Fast}}(g), \qquad (8)$$

where $\Delta^r_{\text{AER-local}}(g)$ is the change in the $r$th stochastically-regionalized $SO_2$ emissions trajectory in grid cell $g$ for 1900 vs. 1966 and $\widehat{\beta}_{\text{Fast}}(g)$ are

from Eq. (5). The reference years 1900 and 1966 are chosen since these are the years in which $SO_2$ emissions were at their minimum and maximum, respectively, in the time period we are analyzing. Uncertainty assessments and stippling are based on applying Eqs. (6)–(8) to resampling-based estimates of these quantities; see ref. 26.

The signal emergence plots in Fig. 3 also show the effect of each forcing agent on precipitation but now over time and aggregated spatially. When assessing temporal changes, we can now explicitly separate trajectories over time due to GHGs, AER-glob, and AER-local. For a given collection of 0.25° × 0.25° grid cells $\mathcal{A}$, we compute area-averaged anomalies in year $t$ using

$$\widehat{P}^{\text{Obs}}_{\text{GHG}}(t) = \frac{1}{\|\mathcal{A}\|}\sum_{g\in\mathcal{A}} a(g)\widehat{\beta}_{\text{Slow}}(g)\left[F_{\text{GHG}}(t,\tau_{\text{Slow}}) - F_{\text{GHG}}(\text{pi-clim},\tau_{\text{Slow}})\right]$$

$$\widehat{P}^{\text{Obs}}_{\text{AER-glob}}(t) = \frac{1}{\|\mathcal{A}\|}\sum_{g\in\mathcal{A}} a(g)\widehat{\beta}_{\text{Slow}}(g)\left[F_{\text{AER-glob}}(t,\tau_{\text{Slow}}) - F_{\text{AER-glob}}(\text{pi-clim},\tau_{\text{Slow}})\right]$$

$$\widehat{P}^{\text{Obs}}_{\text{AER-local}}(t) = \frac{1}{\|\mathcal{A}\|}\sum_{g\in\mathcal{A}} a(g)\widehat{\beta}_{\text{Fast}}(g)\left[F_{\text{AER-local}}(t,\tau_{\text{Fast}},g) - F_{\text{AER-local}}(\text{pi-clim},\tau_{\text{Fast}},g)\right],$$

$$(9)$$

where $a(g)$ is the area of grid cell $g$, $\|\mathcal{A}\| = \sum_{g\in\mathcal{A}} a(g)$, and "pi-clim" refers to a pre-industrial climate (represented by the 1900-1929 average). For brevity, we omit the dependence of the attribution coefficient estimates $\widehat{\beta}_{(\cdot)}$ on forcing trajectory, but note that the above calculations involve (weighted) averages of trajectory-specific estimates and forcing anomalies similar to, e.g., Eq. (6). The sum-total anthropogenic forcing shown in Fig. 3, denoted ANT, is simply

$$\widehat{P}^{\text{Obs}}_{\text{ANT}}(t) = \widehat{P}^{\text{Obs}}_{\text{GHG}}(t) + \widehat{P}^{\text{Obs}}_{\text{AER-glob}}(t) + \widehat{P}^{\text{Obs}}_{\text{AER-local}}(t). \qquad (10)$$

Basic bootstrap confidence intervals are calculated by applying Eq. (9) to bootstrap estimates of the forcing coefficients.

### Analysis of the CMIP6 historical multimodel ensemble

To compare our GHCN analysis with corresponding estimates from Global Climate Models (GCM), we apply the D&A formula described by Eq. (2) to individual ensemble members of each GCM in the CMIP6 historical experiment[58], matching the period covered by the in situ records (i.e., 1900-present). Note that Eq. (2) is applicable to the GCMs precisely because all hypotheses enumerated in Supplementary Table 1 were tested using the CMIP6 multimodel ensemble. The experimental protocol for these simulations prescribes external forcing agents that correspond to the historical period and hence the forcing time series $F_{\text{Slow}}$ and $F_{\text{Fast}}$ are as in Supplementary Fig. 2 (but with GCM-specific values of $\tau_{\text{Slow}}$). Since the historical runs are fully coupled, each ensemble member has its own set of driver time series: these are calculated via the Climate Variability Diagnostics Package [for everything except ELI;[69]] and the Toolkit for Extreme Climate Analysis [for ELI;[70]]. Our analysis of the historical ensemble members mirrors the GHCN analysis but without the use of spatial statistical methods: we simply (1) obtain maximum likelihood estimates of all unknown statistical parameters at each model grid cell and (2) utilize resampling methods to quantify uncertainty. After conducting these analyses, we are left with coefficient estimates $\beta_{\text{Slow}}$ and $\beta_{\text{Fast}}$ as well as measures of uncertainty at each model grid cell and each ensemble member. To explore CONUS-wide changes, we then calculate area-weighted averages of all coefficient estimates and their uncertainties, denoted

$$\left\{\beta_f(j,m), \sigma_f(j,m) : f \in \{\text{Slow, Fast}\}, j = 1,\ldots,n_m\right\}, \qquad (11)$$

where $m = 1,\ldots,M$ indexes climate models and $j = 1,\ldots,n_m$ indexes the ensemble members from model $m$ (for brevity we again omit the dependence of the attribution coefficient estimates $\widehat{\beta}_{(\cdot)}$ on forcing

trajectory). Recall that these quantities are calculated separately for each season and precipitation type (mean and extreme). As with the GHCN analysis, for plotting we convert the coefficient estimates and uncertainties to a precipitation response following Eqs. (6)–(8).

**Model weighting.** Our philosophy for combining estimates of $\beta_{\text{Slow}}$ and $\beta_{\text{Fast}}$ and their effect on seasonal precipitation across the CMIP6 historical multimodel ensemble emphasizes internal consistency of climate models for deriving weights:

1. Down-weight ensemble members $j$ from a given model $m$ with values of $\beta_{\text{Slow}}$ and $\beta_{\text{Fast}}$ that could arise purely by chance, due to internal variability [as quantified by comparing with estimates from CMIP6 Diagnosis, Evaluation, and Characterization of Klima (DECK) pre-industrial control runs;[58]], and

2. Up-weight ensemble members $j$ from a given model $m$ with values of $\beta_{\text{Slow}}$ that are consistent with estimates from corresponding runs of transient $CO_2$-only forcing [i.e., the CMIP6 DECK 1pctCO2 runs;[58]].

All comparisons are made within-model, and note that we do not attempt to specify which climate models are "better" or "worse" with respect to observations. Furthermore, this approach to model weighting emphasizes trends (as opposed to mean climatologies) which is most relevant for this exercise.

**Pre-industrial control (piControl) fitting.** To deprecate "false positive" detection of trends due to anthropogenic forcing agents, we apply Eq. (2) to $i = 1,\ldots,N$ overlapping 121-year segments of pre-industrial control runs (i.e., esm-piControl and/or piControl experiments) from each model (to mirror the length of the GHCN observational record). In each of these fits, we use the observed Slow and Fast forcing time series (from the historical 1900-2020 period; we again use stochastically-regionalized $SO_2$ emissions) but drivers corresponding to the realized conditions in each ensemble member. As with the historical estimates, we apply the formula in each model grid cell and then obtain area-weighted CONUS averages. This procedure detects false positives since the anthropogenic forcing agents are not actually present in the simulations; the piControl estimates reveal the magnitude of secular 121-year trends that can arise from internal climate variability. Let $\{\beta_{PI,f}(i,m), \sigma_{PI,f}(i,m)\}$ represent pairs of area-weighted, CONUS-averaged coefficient estimates and bootstrap standard errors for forcing $f$ and segment $i = 1,\ldots,N$ of model $m$. Then, we calculate a single effect (with uncertainty) for each model as:

$$\beta_{PI,f}(m) = \frac{1}{N}\sum_{i=1}^{N}\beta_{PI,f}(i,m), \quad \sigma_{PI,f}(m) = \sqrt{\frac{1}{N^2}\sum_{i=1}^{N}[\sigma_{PI,f}(i,m)]^2}. \qquad (12)$$

Note that our inclusion criterion is that the esm-piControl and/or piControl runs have at least 500 years of data; we then select the last 500 years of each run, which allows us to fit $N = 19$ overlapping segments of 121 years.

The first component of the model ensemble weights is then one minus the inverse squared exponential of the standardized difference between the piControl estimates (which have arisen purely by chance, from Eq. (12)) and the historical estimates (from Eq. (11)):

$$w_{PI,f}(j,m) = 1 - \exp\{-Z_{PI,f}(j,m)^2\}, \qquad (13)$$

where

$$Z_{PI,f}(j,m) = \frac{\beta_f(j,m) - \beta_{PI,f}(m)}{\sqrt{\sigma_f(j,m)^2 + \sigma_{PI,f}(m)^2}}. \quad (14)$$

**Transient 1pctCO2 fitting.** Next, we prioritize climate models with historical ensemble members that are internally consistent with Slow coefficients estimated from transient $CO_2$-only runs. Here, we again apply Eq. (2) to 1pctCO2 runs (again applied to each grid cell followed by calculating area-weighted averages over CONUS), now considering the initialized ensemble members up through the year in which the lagged forcing from increased $CO_2$ equals the lagged forcing from the collection of GHGs in the historical run, using the lag time-constant $\tau_{Slow}$ specific to each GCM. In the fitting formula, the Slow forcing time series $F_{Slow}(t, \tau_{GHG})$ is set to be the lagged $CO_2$-only forcing (i.e., the AER-glob forcing is omitted since these runs have no anthropogenic aerosols), the Fast forcing $F_{Fast}(t, \tau_{Fast})$ and $\beta_{Fast}$ are set to 0, and the drivers correspond to the real conditions within each ensemble member. This fitting procedure yields $\{\beta_{1\%}(m), \sigma_{1\%}(m)\}$ for each model. For climate models that provide data for more than one ensemble member, we average over the ensemble members as in Eq. (12). The second component of the model ensemble weights is then

$$w_{1\%}(j,m) = \exp\{-Z_{1\%}(j,m)^2\}, \quad (15)$$

where

$$Z_{1\%}(j,m) = \frac{\beta_{Slow}(j,m) - \beta_{1\%}(m)}{\sqrt{\sigma_{Slow}(j,m)^2 + \sigma_{1\%}(m)^2}}. \quad (16)$$

**Model ensemble weighting.** Combining all of the above, we obtain a weight for each ensemble member $j = 1,...,n_m$ of the model $m = 1,...,M$:

$$w(j,m) = \nu\, w_{1\%}(j,m) \prod_f w_{PI,f}(j,m) \text{ s.t. } \sum_m \frac{1}{n_m} \sum_j w(j,m) = 1 \quad (17)$$

where $\nu$ is a normalization factor that enforces the right-hand condition (recall $f \in \{\text{Fast,Slow}\}$). Note that including the $\frac{1}{n_m}$ in between the summations gives each model equal weight; again, recall that these weights are calculated separately for each season and precipitation type (mean and extreme). The number of ensemble members we have for each of the historical, 1pctCO2, and piControl experiments are shown in Supplementary Table 2. A total of $M = 25$ climate models have the requisite data for these three experiments; across these models, we have 316 historical ensemble members (i.e., $\sum_{m=1}^{M} n_m = 316$). Individual weights ($w_{PI,f}(j,m)$ and $w_{1\%}(j,m)$) are plotted against corresponding estimates of each forcing's influence on precipitation (see Eqs. (18) and (19)) in Supplementary Fig. 9. The overall (non-normalized) weights $w(j,m)$ are shown in Supplementary Fig. 10.

**Comparing weighted climate models with observations.** To summarize fits from the individual historical ensemble members, we convert the historical coefficient estimates into their effect on precipitation, averaged over CONUS, as follows. First, in each season and for each precipitation type, the slow precipitation response for ensemble member $j$ of model $m$ is

$$P_{Slow}(j,m) = \Delta_{Slow}(m) \frac{\sum_{g=1}^{G_m} a_m(g)\beta_{Slow}(j,m,g)}{\sum_{g=1}^{G_m} a_m(g)}, \quad (18)$$

where $\Delta_{Slow}(m)$ is the change in lagged GHG and AER-glob forcing for 1900 vs. 2014 for model $m$ (this quantity is model-specific since each model has its own lag $\tau_{Slow}$), $a_m(g)$ is the area of grid cell $g = 1,...,G_m$ of

model $m$, and $\beta_{Slow}(j, m, g)$ is the Slow coefficient estimate in each grid cell $g$. The reference years 1900 and 2014 are chosen since these are when the lagged GHG and AER-glob forcing is at its minimum and maximum, respectively, in the time period bounded by the start of the GHCN record and end of the CMIP6 historical simulations. Next, again in each season and for each precipitation type,

$$P_{Fast}(j,m) = \frac{\sum_{g=1}^{G_m} \Delta_{Fast}(m,g) a_m(g)\beta_{Fast}(j,m,g)}{\sum_{g=1}^{G_m} a_m(g)}, \quad (19)$$

where $\Delta_{Fast}(m, g)$ is the change in stochastically-regionalized $SO_2$ emissions in the grid cell $g$ of the model $m$ for 1900 vs. 1966. Note that $\Delta_{Fast}(m, g)$ also depends on the season. The reference years 1900 and 1966 are chosen since these are the years in which $SO_2$ emissions were at their minimum and maximum, respectively, in the period of interest. For each season and precipitation type across all ensemble members, we obtain the best estimates, a lower 90% confidence bound, and an upper 90% confidence bound, denoted

$$\left\{ \widehat{P}_f(j,m), \widehat{l}_f(j,m), \widehat{u}_f(j,m) \right\}, \quad (20)$$

of the effect on precipitation for $f \in \{\text{Slow,Fast}\}$ after applying Eqs. (18) and (19) to best estimates and bootstrap estimates of each coefficient.

In light of the large number of ensemble members we need to summarize (across seasons, precipitation type, and forcing), we define

$$h_f(x) = \sum_{m=1}^{M} \frac{1}{n_m} \sum_{j=1}^{n_m} w(j,m) \times I(x - \widehat{l}_f(j,m)) \times I(\widehat{u}_f(j,m) - x) \quad (21)$$

(notation for season and precipitation type are suppressed), where

$$I(z) = \begin{cases} 1 & \text{if } z \geq 0 \\ 0 & \text{otherwise.} \end{cases} \quad (22)$$

The function $h_f(x)$ summarizes the weighted proportion of ensemble members for which $x$ is included in their 90% confidence interval and ranges between 0 and 1 by construction. Here $x$ is the anthropogenically forced change in precipitation rate caused by the forcing agent(s) $f$. Given this concise summary of the model results across seasons for each precipitation type, we can plot $h_f(x)$ for a range of $x$ values and compare the resulting curve with observational estimates corresponding to Eq. (20) obtained from the GHCN analysis, denoted

$$\left\{ \widehat{P}_f(\text{GHCN}), \widehat{l}_f(\text{GHCN}), \widehat{u}_f(\text{GHCN}) \right\}. \quad (23)$$

Figure 5 shows our results, comparing fits obtained from the CMIP6-historical multimodel ensemble with those from the in situ records in the GHCN. Note that we have applied a spline-smoothing to the $h_f(\cdot)$ curves for visual appeal.

## Fast versus slow precipitation response to aerosols

The analysis in ref. 43 presents a multimodel assessment of the fast and slow precipitation response to individual climate forcings, including sulfate aerosols ($SO_4$). In the main text, their Fig. 3 shows geographical patterns of multimodel mean precipitation change (fast, slow, and total), while Fig. 4 shows the same quantities for various spatial aggregations, including continental averages. However, their analysis only looked at annual mean precipitation, and furthermore from their Fig. 3 it appears that the North America land average summarized in Fig. 4 is heavily influenced by the precipitation responses in Canada, which is of course not included in our CONUS-specific analysis. To explicitly evaluate the geographical patterns and CONUS-wide means of *seasonal* mean and extreme precipitation responses, we repeat the analysis described in ref. 43. Output from four modeling experiments

is utilized, all of which are part of the Precipitation Driver and Response Model Intercomparison Project [PDRMIP;[57]]:

1. ʙase-fsst: the external forcings correspond to present-day conditions and the sea-surface temperatures are fixed.
2. ꜱulx5-fsst: the external forcings correspond to present-day conditions *except* for sulfate aerosol concentrations, which are multiplied by a factor of 5, and the sea-surface temperatures are fixed.
3. ʙase-coupled: the external forcings correspond to present-day conditions and the sea-surface temperatures are prognostic.
4. ꜱulx5-coupled: the external forcings correspond to present-day conditions *except* for sulfate aerosol concentrations, which are multiplied by a factor of 5, and the sea-surface temperatures are prognostic.

The eight GCMs and number of simulated years used are given in Supplementary Table 3. Following ref. 43, we utilize the last 10 years of the ꜰsst experiments and the last 50 years of the ᴄoupled experiments; we calculate the seasonal mean daily precipitation and seasonal maximum daily precipitation (Rx1Day) in each grid box of each model-experiment. The fast precipitation response is calculated as the mean of ꜱulx5-fsst minus the mean of ʙase-fsst; the total precipitation response is calculated as the mean of ꜱulx5-coupled minus the mean of ʙase-coupled; and the slow precipitation response is the total precipitation response minus the fast precipitation response. To assess geographical patterns of change, all model results are conservatively remapped to a $1° \times 1°$ longitude-latitude grid. For all quantities we calculate a 90% bootstrap confidence interval to summarize uncertainty. Our results are shown in Supplementary Fig. 5.

## Data availability

All global climate data analyzed in this study are available in the Earth System Grid Federation repository, accessible at https://esgf-node.llnl.gov/projects/esgf-llnl/. The in situ precipitation records supporting this article are based on publicly available measurements from the National Centers for Environmental Information (https://www.ncei.noaa.gov/products/land-based-station/global-historical-climatology-network-daily).

## Code availability

All data analysis in this manuscript was conducted using open-source programming languages and software (namely, R and Python). The primary results utilize functionality from the climextRemes[72] and convoSPAT[73] software packages for R.

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

## Acknowledgements

We acknowledge the World Climate Research Programme, which, through its Working Group on Coupled Modelling, coordinated and promoted CMIP6. We thank the climate modeling groups for producing and making available their model output, the Earth System Grid Federation (ESGF) for archiving the data and providing access, and the multiple funding agencies who support CMIP6 and ESGF. This research was supported by the Director, Office of Science, Office of Biological and Environmental Research of the U.S. Department of Energy under Contract No. DE-AC02-05CH11231 and by the Regional and Global Model Analysis Program area within the Earth and Environmental Systems Modeling Program (MDR, WDC, MFW, TAO, HH, PAU). The research used resources of the National Energy Research Scientific Computing Center (NERSC), also supported by the Office of Science of the U.S. Department of Energy, under Contract No. DE-AC02-05CH11231. This project was supported by the Environmental Resilience Institute, funded by Indiana University's Prepared for Environmental Change Grand Challenge initiative (TAO). Work performed by PAU is partially under the auspices of the U.S. Department of Energy by Lawrence Livermore National Laboratory under Contract DE-AC52-07NA27344. This document was prepared as an account of work sponsored by the United States Government. While this document is believed to contain correct information, neither the United States Government nor any agency thereof, nor the Regents of the University of California, nor any of their employees, makes any warranty, express or implied, or assumes any legal responsibility for the accuracy, completeness, or usefulness of any information, apparatus, product, or process disclosed, or represents that its use would not infringe privately owned rights. Reference herein to any specific commercial product, process, or service by its trade name, trademark, manufacturer, or otherwise, does not necessarily constitute or imply its endorsement, recommendation, or favoring by the United States Government or any agency thereof, or the Regents of the University of California. The views and opinions of the authors expressed herein do not necessarily state or reflect those of the United States Government or any agency thereof or the Regents of the University of California.

## Author contributions

M.D.R. and W.D.C. contributed initial conception of the study and led all analyses, figure generation, and writing. M.F.W. contributed interpretation of results and comparison between observed and simulated changes. T.A.O. contributed formal analysis (application of TECA to calculate ELI), funding acquisition, and software development (the TECA ELI algorithm). HH contributed to data interpretation and manuscript writing. P.A.U. contributed conservative data regridding and translations from the original model data, necessary for intercomparison. M.D.R., W.D.C., M.F.W., T.A.O., H.H., and P.A.U. contributed to the interpretation of study results and editing of the study's writing and figures.

## Competing interests

The authors declare no competing interests.
