## [Peer Review File · Nature Communications]

Anthropogenic aerosols mask increases in US rainfall by greenhouse gasesEditorial Note: This manuscript has been previously reviewed at another journal that is not operating a transparent peer review scheme. This document only contains reviewer comments and rebuttal letters for versions considered at *Nature Communications*.

Reviewers' comments:

Reviewer #1 (Remarks to the Author):

Review of the revised version of "Anthropogenic aerosols mask increases in US rainfall by greenhouse gases" for Nature Communications (initial submission to Nature)

#reviewer 1

I appreciate the authors' responding to my comments, including both major ones on interpretation and minor ones on language etc. The revisions made together with the explanations in the response to reviewers alleviates the points I raised in the previous review. Besides the new text in 318ff and 499ff I also appreciate some of the changes made in response to the other reviewers, for instance, the "Comparison with optimal fingerprinting methods" section in the appendix and what is now Fig. 1, and think they improve the manuscript. The following few minor points remain.

Comment previous review

- Fig. 2 what does the linewidth of the dashed lines show? > All dashed lines are the same width | Yes, sorry, that seems to be the way okular shows pdf figures, depending on how much I zoom in, some lines are thinner and thicker.

Minor points new text

- I find "equal and opposite [aerosol and greenhouse gas signals]" super confusing, at least as non-native English speaker, does it mean "equal (in magnitude) and opposite (in sign)" (aerosols vs ghg) or "same sign (in some season) and different sign (in other seasons)" aerosols vs ghg or a third meaning I'm not getting? If there is a way to clarify this in the text (I'm aware of the abstract's word limit), that would be good.
- these emission sources >each< increase or decrease rainfall?

- 60-61 "meaningful", unclear what that means (having impacts? statistically significant? detectable?)

- 66-72 This new argument needs to be better linked. Firstly, I think this "even" is opaque, especially since in line 67 "aerosol" is the first time mentioned in the text (discounting the abstract), so how ant. aerosol and human influence are linked has not been introduced. Secondly I disagree with the logic when comparing the SNR from individual simulations with that for the multi-model mean, since both internal variability and model uncertainty play a role in that comparison, so from this alone you can't conclude on the model uncertainty contribution I think.

- 111 what does this "otherwise" mean?

- 153 I'd remove "here" since the reader might not be in the conus...

- 154 characterise so₂ – is very broad a term, I think you mean characterising some specific aspects of so₂ that are relevant for its influence on climate, perhaps consider specifying.

- 162 sounds a bit weird since something that is prescribed IMO doesn't count as "showing agreement across" models, perhaps rephrase

- 232, 237, 238 "vs" here I find quite confusing

- Tab. 1. Good idea to synthesise this. Though if I understand this correctly, this now includes any grid boxes, regardless of whether the statistical attribution is significant. Must be, since the significance hatching differs for aerosols and GHG grid boxes. I suggest it would be more meaningful if the authors could add a third option for each AER and GHG so it's "- | + | nonsignificant" or "- | + | <0.3mm/day" (or 0.1 at least, what's white in Fig. 2. Then I also suggest to make the diagonal entries (-/-, +/+, n/n) in bold or perhaps rather the ones where the signals clearly counteract (-/+ and +/-) to make the table easier to read again...

- 354 typo precipitaiton

- 424-428 similar point as above to 66-72: does the broad range of individual ensemble members really illustrate the broad range in responses to anthropogenic forcing meaning that individual climate models [not: climate model runs?] cannot confidently attribute [...]?

431 "wrong" seems a strong word here, if "right" is the result derived w/ your forcing-only (rather than model) based method? perhaps specify instead what you are comparing to, or use a more neutral term like "different"

Kind regards, Sabine Undorf

Reviewer #2 (Remarks to the Author):

Thanks for the responses from the authors, which considerably improved the manuscript. But I don't think the authors completely addressed my previous major comment #1 on the remote aerosol effect and comment #2 on the fast vs slow responses. These two hypotheses basically constitute the foundation of their framework. I don't think this manuscript can be accepted in Nature Communication until they are well justified.

1. To demonstrate the marginal effect of remote aerosol on the CONUS precipitation, the author generated Fig. B2 showing that the CONUS-average responses to the Asian aerosols are statistically

indistinguishable from zero. Thanks for the work here but I still have a couple of remaining questions here.

(a) The authors mainly use the bootstrap method and permutation approach (for spatial pattern) to estimate the uncertainty based on their published paper. I think the authors should briefly summarize how they conducted these two methods and why they fit to make the reader understand the contents easily rather than refer to their paper to learn the complicated statistical contents in much detail. For instance, how the bootstrap is conducted for Fig. B.2.

(b) The authors just show the continental-scale precipitation mean changes. What about the spatial distribution of precipitation responses to aerosols? There could be strong increases and decreases on a regional scale. Averaging these anomalies could smooth out the aerosol signals.

(c) As the authors pointed out in this study, the aerosol influences on precipitation are in comparable magnitude to that of GHGs. The changes indistinguishable for zero don't necessarily mean that the effects of remote aerosols are not important. It may be only because of the large uncertainty in the presentation of aerosol impact in climate models. For example, the changes in HadGEM3 are not marginal to me, as well as in some seasons in other models. Discussion on this should also be made.

(d) I understand that the Asian aerosol level is low during 1900–1966, which may not affect the detection and attribution strongly in this work. Instead, the European aerosols could be important in affecting the CONUS precipitation. Looking at the influence of European aerosols (e.g., black carbon and sulfate) on the CONUS precipitation is essential.

2. For the fast vs slow responses, the authors mentioned that the argument in Richardson et al. (2018) supports the hypothesis that the sulfate aerosol-induced fast precipitation changes dominate on the global land scale, which is however not cited in the revised manuscript. I agree that fast responses are more important than slow responses in land-mean precipitation changes supported by Fig. 4 in Samset et al. (2016) and Fig. 8a Richardson et al. (2018). But this argument may vary regionally as seen in Fig. 4 in Samset et al. (2016) (as demonstrated by the authors also), particularly in the CONUS, where is the focus of this study. The slow responses are far more important than fast responses in the North America (NAM) region. Again, this means the authors probably have to consider the lagged effects of sulfate aerosols rather than omit them.

Specific comments:

L242: please provide details on how you define 20-year return values of extreme daily precipitation.

Reference

Richardson, T. B., Forster, P. M., Andrews, T., Boucher, O., Faluvegi, G., Fläschner, D., et al. (2018). Drivers of precipitation change: An energetic understanding. *Journal of Climate*. <https://doi.org/10.1175/JCLI-D-17-0240.1>

Samset, B. H., Myhre, G., Forster, P. M., Hodnebrog, Andrews, T., Faluvegi, G., et al. (2016). Fast and slow precipitation responses to individual climate forcings: A PDRMIP multimodel study. *Geophysical Research Letters*. <https://doi.org/10.1002/2016GL068064>

Reviewer #3 (Remarks to the Author):

I appreciate the authors' effort to consider my previous comments. Although the revised manuscript provides more explanations and arguments, unfortunately, I find my major concerns not addressed clearly.

First, authors' intentional choice of not using GCMs to estimate fingerprints would help reduce uncertainties in AER fingerprints, but directly linking SO₂ emissions with local precipitation is not supported well by physical processes, implying the large uncertainty remaining in their 'attribution' results to AER. In addition, authors describe similarities and differences of their d&a method compared to traditional d&a methods (OF). This helps to understand the authors' method, but the main question about how 'noise' of scaling factors is estimated and boosts S/N remains unclear. As I understand from their responses, authors seem to use weather variability (P_W) estimated from the observations as a major component of noise but OF usually considers P_D (its long-term variations). This indicates that the proposed d&a method will have larger noise than OF and therefore increases S/N by ignoring signal uncertainties due to model differences (i.e. using forcing data to estimate signal) rather than by reducing noise.

Second, main conclusions "AER masks rainfall increase by GHG in US" remain not robust. Responding to reviewer #1's concerns, authors removed "resolve outstanding uncertainties regarding the human influence on regional precipitation" or similar interpretations throughout their manuscript (except Abstract?). This clarifies that this study does not advance our understanding of 'regional scale' d&a of precipitation changes, which is suggested as the aim. Also, authors still apply different periods when attributing the observed changes to GHG and AER (Figures 1, 2, and 4, and Table 1 as well) such that 1900-1966 trends are used for AER and 1900-2020 trends for GHG. Before and after 1966, we can expect difference contribution by AER as reviewer #1 pointed. Under this different timing, how can they conclude that AER mask GHG influences? Their important motivation is the large uncertainty in Figure SPM.3 of IPCC AR6 but that figure considered the observed trends since 1950. Thus, results from this study cannot be compared properly with the IPCC results.

Lastly, another important finding was suggested as attribution at local scales, particularly AER influences as the title says. However, major conclusions are obtained based on US averages and local scale D&A is found very limitedly in space and time. They even say that "The AER signal is essentially non-existent outside JJA when considering spatial scales smaller than CONUS." Given the limited impact of AER (aside from the different period issue), how can we conclude that "AER signal masks GHG signal in US"? In terms of physical mechanisms for AER attribution in JJA, authors only speculate that "convective invigoration by SO₂" may have worked, without showing clear evidences.

Some specific points:

L28-30: "Aerosol emissions offset these increases in the winter and spring but appear to enhance rainfall during the summer and fall." This will be dependent on periods considered. I think authors need to consider different periods and quantify GHD and AER contributions to the observed trends to clarify this issue.

L74-76: Proposed method does not “disentangle the complex causes of regional precipitation change”.

L84-87: “this decomposition allows us to conclusively attribute changes to these forcing agents” This decomposition simply estimates contribution from each forcing, not conclusively attributing “observed” change to “each” forcing.

L381-385: “In summary, our results show that uncertainties related to the emergence of a detectable and attributable human influence on regional precipitation over the CONUS over the historical record [Figure SPM.3; 31] can in most cases be explained by decomposing the combined ANT signal into its two components, GHG and AER forcing.” Results do not support this solid conclusion for regional attribution. Also, Figure SPM3 considers heavy precipitation trends since 1950 but the authors’ analysis covers different periods, particularly for AER.

Line 617-620: “we quantify uncertainty via resampling methods, specifically a block bootstrap for estimating standard errors and confidence intervals [as in 69] and a permutation/reshuffling approach for ascribing statistical significance to spatial patterns [as in 26].” This is the important procedure for estimating uncertainty in scaling factors so authors need to explain details in comparison with traditional d&a methods. How does this consider internal variability, particularly P_D and P_W?

L630-631: “the proportion of type I errors at 0.33 (for moderate significance) and 0.1 (for strong significance)”. Figure 2 shows that grid-box attribution results for AER have ‘moderate’ significance only in all cases (almost no grid shows ‘strong’ significance), indicating AER signal is not detectable when applying 5% level as used in OF. Is there any specific reason to use the lower significance level?

Line 676-679: “Therefore, in our case the D&A exercise is reduced to assessing a single null hypothesis test (i.e., rejecting $H_0: \beta_f = 0$ both detects and attributes forcing f), such that larger β_f (in absolute value) implies a stronger influence of forcing f .” I think this is equivalent to the ‘detection’ of signals in OF. Can authors think about another hypothesis test for ‘attribution’ by which they check agreement of the forced trend with the observed trend. I think this sort of quantification of forcing contribution is needed for robust ‘attribution’.

**Comments from Reviewer #1**
I appreciate the authors' responding to my comments, including both major ones on interpretation and minor
ones on language etc. The revisions made together with the explanations in the response to reviewers allevi-
ate the points I raised in the previous review. Besides the new text in 318ff and 499ff I also appreciate some
of the changes made in response to the other reviewers, for instance, the "Comparison with optimal finger-
printing methods" section in the appendix and what is now Fig. 1, and think they improve the manuscript.
The following few minor points remain.
Thank you for your continued review of our manuscript!
*Comment previous review*
• Fig. 2 what does the line width of the dashed lines show? "All dashed lines are the same width" –
Yes, sorry, that seems to be the way Okular shows .pdf figures, depending on how much I zoom in,
some lines are thinner and thicker.
Thank you! Glad this was just an aspect of the PDF viewer.
*Minor points new text*
• 32 I find "equal and opposite [aerosol and greenhouse gas signals]" super confusing, at least as a
non-native English speaker, does it mean "equal (in magnitude) and opposite (in sign)" (aerosols vs
GHGs) or "same sign (in some season) and different sign (in other seasons)" aerosols vs GHGs or a
third meaning I'm not getting? If there is a way to clarify this in the text (I'm aware of the abstract's
word limit), that would be good.
Excellent point, and our apologies for the unclear English. Our intention is your first interpretation:
equal in magnitude and opposite in sign. We have replaced "equal and opposite" with "offsetting,"
which we hope is more clear.
• 35 these emission sources each increase or decrease rainfall?
Correct – GCMs are unclear on whether each forcing individually increases or decreases rainfall. We
have changed this sentence to "whether each emissions source increases or decreases rainfall."
• 60-61 "meaningful", unclear what that means (having impacts? statistically significant? detectable?)
Their study implied statistical significance – we have changed "meaningful" to "statistically signifi-
cant."
• 66-72 This new argument needs to be better linked. Firstly, I think this "even" is opaque, especially
since in line 67 "aerosol" is the first time mentioned in the text (discounting the abstract), so how ant.
aerosol and human influence are linked has not been introduced. Secondly, I disagree with the logic
when comparing the SNR from individual simulations with that for the multimodel mean, since both
internal variability and model uncertainty play a role in that comparison, so from this alone you can't
conclude on the model uncertainty contribution, I think.
Thank you for raising this point. We have now noted the importance of anthropogenic aerosols and
 focused the argument on the behavior of individual ensemble members; see lines 69-74.
• 111 what does this “otherwise” mean?
 Good point – I think the sentence reads better with “otherwise” omitted.
• 153 I’d remove “here” since the reader might not be in the CONUS...
 Great point, done!
• 154 characterize so2 – is very broad a term, I think you mean characterizing some specific aspects of
 so2 that are relevant to its influence on climate, perhaps consider specifying.
 Thanks for raising this point – replaced with “the effects of SO₂ on precipitation”.
• 162 sounds a bit weird since something that is prescribed IMO doesn’t count as “showing agreement
 across” models, perhaps rephrase
 Good point. Replaced with “Aerosol emissions are a prescribed quantity in historical simulations and
 are hence consistent across climate models, ...”
• 232, 237, 238 “vs” here I find quite confusing
 Our apologies. This text has been removed and rephrased to talk about fast vs. slow precipitation
 responses.
• Tab. 1. Good idea to synthesize this. Though if I understand this correctly, this now includes any grid
 boxes, regardless of whether the statistical attribution is significant. Must be, since the significance
 of hatching differs for aerosols and GHG grid boxes. I suggest it would be more meaningful if
 the authors could add a third option for each AER and GHG, so it’s “□ | + | non-significant” or
 “□ | + | <0.3mm/day” (or 0.1 at least, what’s white in Fig. 2. Then I also suggest making the
 diagonal
 entries (-/-, +/-, n/n) in bold or perhaps rather the ones where the signals clearly counteract (-/+ and
 +/-) to make the table easier to read again...
 Glad you liked the table. Your point is well taken; however, in order to keep the table from being too
 busy (e.g., we’d need to summarize where one of Fast vs. Slow is significant, as well as both), we’ve
 elected to keep the table as is. The new Figure 4 summarizes the overall effect of GHGs and aerosols
 as well as their combined significance.
• 354 typo precipitaiton
 Fixed.
• 424-428 similar point as above to 66-72: does the broad range of individual ensemble members
 really illustrate the broad range in responses to anthropogenic forcing meaning that individual climate
 models [not: climate model runs?] cannot confidently attribute [...]?
 Excellent point. Similar to before, the text has been revised to focus on individual climate models.
• 431 “wrong” seems a strong word here, if “right” is the result derived w/ your forcing-only (rather
 than model) based method? perhaps specify instead what you are comparing to, or use a more neutral
 term like “different”
 We concur that this language is too strong. We have revised this sentence to reflect that in some cases
 GHG-driven changes result in decreases.
Kind regards,
Sabine Undorf
Dear Sabine, thank you again for your extensive and extremely helpful feedback on our manuscript! It has
been made much better as a result of your comments.
**Comments from Reviewer #2**
Thanks for the responses from the authors, which considerably improved the manuscript. But I don't think
the authors completely addressed my previous major comment #1 on the remote aerosol effect and com-
ment #2 on the fast vs slow responses. These two hypotheses basically constitute the foundation of their
framework. I don't think this manuscript can be accepted in Nature Communication until they are well
justified.
We greatly appreciate your continued input and feedback on our manuscript, and we apologize for failing
to address your feedback from the previous review. We refer you to summary of the large changes made
described in our response to the AE above. We feel that the manuscript and results are greatly improved
thanks to your suggestion to model the slow precipitation response to aerosols and hope you do as well.
1. To demonstrate the marginal effect of remote aerosol on the CONUS precipitation, the author gen-
erated Fig. B2 showing that the CONUS-average responses to the Asian aerosols are statistically
indistinguishable from zero. Thanks for the work here, but I still have a couple of remaining ques-
tions here.
(a) The authors mainly use the bootstrap method and permutation approach (for spatial pattern) to
estimate the uncertainty based on their published paper. I think the authors should briefly sum-
marize how they conducted these two methods and why they fit to make the reader understand
the contents easily rather than refer to their paper to learn the complicated statistical contents in
much detail. For instance, how the bootstrap is conducted for Fig. B.2.
(b) The authors just show the continental-scale precipitation mean changes. What about the spatial
distribution of precipitation responses to aerosols? There could be strong increases and decreases
on a regional scale. Averaging these anomalies could smooth out the aerosol signals.
(c) As the authors pointed out in this study, the aerosol influences on precipitation are in comparable
magnitude to that of GHGs. The changes indistinguishable for zero don't necessarily mean that
the effects of remote aerosols are not important. It may be only because of the large uncertainty
in the presentation of aerosol impact in climate models. For example, the changes in HadGEM3
are not marginal to me, as well as in some seasons in other models. Discussion on this should
also be made.
(d) I understand that the Asian aerosol level was low during 1900-1966, which may not affect the
detection and attribution strongly in this work. Instead, the European aerosols could be important
in affecting the CONUS precipitation. Looking at the influence of European aerosols (e.g., black
carbon and sulfate) on the CONUS precipitation is essential.
Given our pivot to incorporate the slow precipitation response to aerosols, we have removed all men-
tion of the Asian aerosol analysis, including Supplemental Figure B.2.
2. For the fast vs slow responses, the authors mentioned that the argument in Richardson et al. (2018)
supports the hypothesis that the sulfate aerosol-induced fast precipitation changes dominate on the
global land scale, which is however not cited in the revised manuscript. I agree that fast responses are
more important than slow responses in land-mean precipitation changes supported by Fig. 4 in Samset
et al. (2016) and Fig. 8a Richardson et al. (2018). But this argument may vary regionally as seen
in Fig. 4 in Samset et al. (2016) (as demonstrated by the authors also), particularly in the CONUS,
where is the focus of this study. The slow responses are far more important than fast responses in the
North America (NAM) region. Again, this means the authors probably have to consider the lagged
effects of sulfate aerosols rather than omit them.
Thank you for this point. As described above, we have now included the lagged effects of sulfate
aerosols. A version of the Samset et al. (2016) annual mean precipitation result (reproduced in our
paper, see “Fast versus slow precipitation response to aerosols” in the Methods and also Supplemental
Figure B.5) verifies that our results are in agreement with modeling experiments from PDRMIP –
especially in the summer, which was something we previously had trouble interpreting.
*Specific comments:*
• L242: please provide details on how you define 20-year return values of extreme daily precipitation.
Thank you for raising this point. We have added a brief description for this calculation in the caption
of Figure 2.
**Comments from Reviewer #3**
I appreciate the authors’ effort to consider my previous comments. Although the revised manuscript provides
more explanations and arguments, unfortunately, I find my major concerns not addressed clearly.
Thank you for your continued review of our manuscript – we greatly appreciate your feedback.
First, authors’ intentional choice of not using GCMs to estimate fingerprints would help reduce uncertainties
in AER fingerprints, but directly linking SO₂ emissions with local precipitation is not supported well by
physical processes, implying the large uncertainty remaining in their ‘attribution’ results to AER. In addition,
authors describe similarities and differences of their d&a method compared to traditional d&a methods (OF).
This helps to understand the authors’ method, but the main question about how ‘noise’ of scaling factors
is estimated and boosts S/N remains unclear. As I understand from their responses, authors seem to use
weather variability (PW) estimated from the observations as a major component of noise but OF usually
considers PD (its long-term variations). This indicates that the proposed d&a method will have larger noise
than OF and therefore increases S/N by ignoring signal uncertainties due to model differences (i.e. using
forcing data to estimate signal) rather than by reducing noise.
Thank you for this comment. Hopefully you are convinced that our inclusion of the slow precipitation
response to aerosols (and favorable comparison with PDRMIP simulations) resolves some of your concerns
with the uncertainty assessments in our attribution statements.
Second, the main conclusions “AER masks rainfall increase by GHG in US” remain not robust. Responding
to reviewer #1’s concerns, the authors removed “resolve outstanding uncertainties regarding the human influ-
ence on regional precipitation” or similar interpretations throughout their manuscript (except the Abstract?).
This clarifies that this study does not advance our understanding of ‘regional scale’ d&a of precipitation
changes, which is suggested as the aim. Also, authors still apply different periods when attributing the ob-
served changes to GHG and AER (Figures 1, 2, and 4, and Table 1 as well) such that 1900–1966 trends are
used for AER and 1900–2020 trends for GHG. Before and after 1966, we can expect difference contribution
by AER, as reviewer #1 pointed. Under this different timing, how can they conclude that AER mask GHG
influences? Their important motivation is the large uncertainty in Figure SPM.3 of IPCC AR6, but that fig-
ure considered the observed trends since 1950. Thus, results from this study cannot be compared properly
with the IPCC results.
Thank you for this feedback, and we apologize for what seems to be an ongoing confusion about the inter-
pretation of the attribution coefficients, β_{Fast} and β_{Slow} . All of our results (including Figures 1, 2, and 4, and
Table 1) are based on estimates of the attribution coefficients obtained from analyzing the entire 121-year
record. As such, there is no difference in time period for the AER trends vs. the GHG trends. The 1900-
1966 for AER and 1900-2020 for GHG time periods are simply a way to rescale the attribution coefficients
so that (1) they are in sensible/interpretable units (mm/day) and (2) we can compare the maximum effect
of each forcing agent in units that make sense. We have changed various things to drive this point home,
e.g., the label on the color bar in Figure 2 is “Change in precipitation (mm/day).” Also, the new Figure 4
shows how the aerosol masking impacts GHG trends at a variety of spatial scales, from CONUS down to
H 0.5Mm².
Lastly, another important finding was suggested as attribution at local scales, particularly AER influences as
the title says. However, major conclusions are obtained based on US averages and local scale D&A is found
very limited in space and time. They even say that “The AER signal is essentially non-existent outside JJA
when considering spatial scales smaller than CONUS.” Given the limited impact of AER (aside from the
different period issue), how can we conclude that “AER signal masks GHG signal in the US”? In terms of
physical mechanisms for AER attribution in JJA, authors only speculate that “convective invigoration by
SO₂” may have worked, without showing clear evidence.
Again, we refer you to the new Figure 4 and discussion on lines 367ff regarding both attribution and masking
at local scales. Additionally, the PDRMIP comparison of fast precipitation responses gives credence to our
observational results.
Some specific points:
• L28-30: “Aerosol emissions offset these increases in the winter and spring but appear to enhance
rainfall during the summer and fall.” This will be dependent on periods considered. I think authors
need to consider different periods and quantify GHG and AER contributions to the observed trends to
clarify this issue.
As clarified above, our conclusions are based on the entire time period, and statements such as the
one in the abstract address the isolated effect of individual forcing agents (answering questions like
“when aerosol emissions increase, does precipitation increase or decrease?”).
• L74-76: Proposed method does not “disentangle the complex causes of regional precipitation change”.
We respectfully disagree, particularly given the new focus of our analysis on fast vs. slow precipitation
responses and the localized conclusions that can be made.
• L84-87: “this decomposition allows us to conclusively attribute changes to these forcing agents”
This decomposition simply estimates contribution from each forcing, not conclusively attributing “ob-
served” change to “each” forcing.
Again, we respectfully disagree – this is exactly what is shown in, e.g., Figure 2.
• L381-385: “In summary, our results show that uncertainties related to the emergence of a detectable
and attributable human influence on regional precipitation over the CONUS over the historical record
[Figure SPM.3; 31] can in most cases be explained by decomposing the combined ANT signal into its
two components, GHG and AER forcing.” Results do not support this solid conclusion for regional
attribution. Also, Figure SPM3 considers heavy precipitation trends since 1950, but the authors’
analysis covers different periods, particularly for AER.
Again, we respectfully disagree. Your point about the IPCC results being 1950-present is well taken;
the IPCC result was more used as evidence that large uncertainties remain in attributing changes to
regional precipitation.
• Line 617-620: “we quantify uncertainty via resampling methods, specifically a block bootstrap for
estimating standard errors and confidence intervals [as in 69] and a permutation/reshuffling approach
for ascribing statistical significance to spatial patterns [as in 26].” This is the important procedure for
estimating uncertainty in scaling factors, so the authors need to explain details in comparison with
traditional d&a methods. How does this consider internal variability, particularly *PD* and *PW*?
Thank you for raising this point, which is well-taken. In this case, we obtain best estimates of the
*PD* and *PW* terms using the “original” (not resampled) data. Resampling methods (bootstrap and
reshuffling) allow us to build a sampling distribution (via bootstrap) and null distribution for testing
(via reshuffling) that explicitly account for internal variability by quantifying the relative magnitude
of year-to-year variability beyond what is described by *PF*.
• L630-631: “the proportion of type I errors at 0.33 (for moderate significance) and 0.1 (for strong
significance)”. Figure 2 shows that grid-box attribution results for AER have ‘moderate’ significance
only in all cases (almost no grid shows ‘strong’ significance), indicating AER signal is not detectable
when applying 5% level as used in OF. Is there any specific reason to use the lower significance level?
Fair point; however, these thresholds correspond to the “likely” and “very likely” language used by
the IPCC and elsewhere. Ultimately, any specific significance level (including, e.g., the ubiquitous
$\alpha = 0.05$) is a subjective choice, for which the evidence against falsifiable hypotheses should always
be judged in light of. Here, we have found it helpful to present multiple significance statements to
help the reader understand the strength of evidence for the claims made throughout the paper.
• Line 676-679: “Therefore, in our case the D&A exercise is reduced to assessing a single null hypoth-
esis test (i.e., rejecting $H_0, f: \beta f = 0$ both detects and attributes forcing f), such that larger βf (in
absolute value) implies a stronger influence of forcing f .” I think this is equivalent to the ‘detection’
of signals in OF. Can authors think about another hypothesis test for ‘attribution’ by which they check
agreement of the forced trend with the observed trend. I think this sort of quantification of forcing
contribution is needed for robust ‘attribution’.
Excellent question. I think this comes back to our broader discussion about the use of GCMs – in
our case, our estimates of the forced trend *are* the observed trends. We would like to point out that
we have a brief mention in the discussion of how one could compare overall trends from observations
with corresponding quantities from the GCMs – see our Supplemental Figure B.14.
References
G. Myhre, P. M. Forster, B. H. Samset, O. Hodnebrog, J. Sillmann, S. G. Aalbergstjø, T. Andrews,
O. Boucher, G. Faluvegi, D. Fläschner, T. Iversen, M. Kasoar, V. Kharin, A. Kirkevåg, J.-F. Lamar-
que, D. Olivié, T. B. Richardson, D. Shindell, K. P. Shine, C. W. Stjern, T. Takemura, A. Voulgarakis,
and F. Zwiers. PDRMIP: A precipitation driver and response model intercomparison project—protocol
and preliminary results. *Bulletin of the American Meteorological Society*, 98(6):1185 – 1198, 2017.
doi:[10.1175/BAMS-D-16-0019.1](https://doi.org/10.1175/BAMS-D-16-0019.1). 2
B. H. Samset, G. Myhre, P. M. Forster, O. Hodnebrog, T. Andrews, G. Faluvegi, D. Fläschner, M. Kasoar,
333 V. Kharin, A. Kirkevåg, J.-F. Lamarque, D. Olivié, T. Richardson, D. Shindell, K. P. Shine, T. Takemura,
and A. Voulgarakis. Fast and slow precipitation responses to individual climate forcings: A PDRMIP
multimodel study. *Geophysical Research Letters*, 43(6):2782–2791, 2016. doi:[10.1002/2016GL068064](https://doi.org/10.1002/2016GL068064).
2, 3
C. Smith. chrisroadmap/aerosol-history: Energy budget constraints on the time history of aerosol forcing,
2021. URL <https://zenodo.org/record/4624765>. 2
C. J. Smith, G. R. Harris, M. D. Palmer, N. Bellouin, W. Collins, G. Myhre, M. Schulz, J.-C. Golaz,
340 M. Ringer, T. Storelvmo, and P. M. Forster. Energy budget constraints on the time history of aerosol
forcing and climate sensitivity. *Journal of Geophysical Research: Atmospheres*, 126(13), July 2021.
doi:[10.1029/2020jd033622](https://doi.org/10.1029/2020jd033622). 2

REVIEWERS' COMMENTS

Reviewer #2 (Remarks to the Author):

I suggest accepting this manuscript published in Nature Communications.

Reviewer #3 (Remarks to the Author):

I am satisfied with the authors' responses to my previous comments and would like to recommend acceptance in the present form. In particular, the separate consideration of fast and slow precipitation response to aerosols as well as the improved explanations of their methods have clarified the major issues.